# A proteomics approach to isolating neuropilin-dependent α5 integrin trafficking pathways: neuropilin 1 and 2 co-traffic α5 integrin through endosomal p120RasGAP to promote polarised fibronectin fibrillogenesis in endothelial cells
Christopher J. Benwell [1,2] ✉, Robert T. Johnson [1,2,3], James A. G. E. Taylor[1,2], Jordi Lambert [1,2,4] & Stephen D. Robinson [1,2] ✉

Integrin trafficking to and from membrane adhesions is a crucial mechanism that dictates many aspects of a cell's behaviour, including motility, polarisation, and invasion. In endothelial cells (ECs), the intracellular traffic of α5 integrin is regulated by both neuropilin 1 (NRP1) and neuropilin 2 (NRP2), yet the redundancies in function between these co-receptors remain unclear. Moreover, the endocytic complexes that participate in NRP-directed traffic remain poorly annotated. Here we identify an important role for the GTPase-activating protein p120RasGAP in ECs, promoting the recycling of α5 integrin from early endosomes. Mechanistically, p120RasGAP enables transit of endocytosed α5 integrin-NRP1-NRP2 complexes to Rab11$^+$ recycling endosomes, promoting cell polarisation and fibronectin (FN) fibrillogenesis. Silencing of both NRP receptors, or p120RasGAP, resulted in the accumulation of α5 integrin in early endosomes, a loss of α5 integrin from surface adhesions, and attenuated EC polarisation. Endothelial-specific deletion of both NRP1 and NRP2 in the postnatal retina recapitulated our in vitro findings, severely impairing FN fibrillogenesis and polarised sprouting. Our data assign an essential role for p120RasGAP during integrin traffic in ECs and support a hypothesis that NRP receptors co-traffic internalised cargoes. Importantly, we utilise comparative proteomics analyses to isolate a comprehensive map of NRP1-dependent and NRP2-dependent α5 integrin interactions in ECs.

Dynamic adhesion to the extracellular matrix (ECM) is primarily mediated by integrin receptors. Blood vessel expansion during vascular development relies on the interaction between the ECM component fibronectin (FN) and its major integrin receptor α5β1 to establish, and subsequently balance the apico-basal polarity of migrating endothelial cells (ECs)[1,2]. During this process, the endocytosis and recycling of active α5β1 integrin receptors (those bound to an FN ligand), facilitates the turnover and replenishment of a newly synthesised matrix. Once the apico-basal axis has been defined, cycles of active α5β1 integrin traffic promote the assembly of a mechano-sensing fibrillar FN network, and

[1]Food Microbiome and Health Programme, Quadram Institute Bioscience, Norwich Research Park, Norwich, UK. [2]School of Biological Sciences, University of East Anglia, Norwich Research Park, Norwich, UK. [3]Present address: Department of Biomedicine, Aarhus University, Aarhus, Denmark. [4]Present address: Section of Cardiorespiratory Medicine, University of Cambridge, VPD Heart & Lung Research Institute, Papworth Road, Cambridge Biomedical Campus, Cambridge, UK. ✉e-mail: christopher.benwell@quadram.ac.uk; stephen.robinson@quadram.ac.uk

therefore preserve a self-sustaining signalling cascade to maintain EC polarity[3].

Depending on integrin heterodimer composition and ECM ligand attachment, integrin traffic involves the transit through specific RabGTPase[+] compartments. For example, whilst α5β1 integrin and αVβ3 integrin are endocytosed collectively via Rab5 or Rab21 GTPases, they are subsequently recycled via diverging Rab4-dependent (αVβ3 integrin) or Rab11-dependent (α5β1 integrin) routes[4–7]. The transmembrane glycoprotein neuropilin 1 (NRP1) was previously reported to selectively promote the endocytosis of FN-bound active α5β1 integrin from disassembling adhesions via its cytosolic interactor GAIP-interacting protein C terminus member 1 (Gipc1) and its associated motor protein myosin 6 in ECs. α5β1 integrin was subsequently shown to recycle in NRP1-associated complexes to stimulate FN fibrillogenesis[8]. We have since revealed that neuropilin 2 (NRP2), the ortholog of NRP1, also promotes Rab11-associated recycling of the α5 integrin subunit in ECs, and regulates the deposition of cell-secreted FN[9,10]. Since genetic ablation of either NRP1 or NRP2 has been shown to impair sprouting angiogenesis[9,11,12], and a simultaneous co-deletion effectively inhibits pathological angiogenesis[13], it has been proposed that a degree of synergy exists between the two receptors. It is unclear, however, which redundancies exist during NRP1-directed and NRP2-directed α5β1 integrin traffic, and which endocytic complexes participate.

p120RasGAP (alias: RASA1) is a ubiquitously expressed endosomal GTPase-activating protein (GAP) that contains canonical phospho-Tyrosine (pTyr)-interacting SH2 domains[14,15]. As an important negative regulator of the Ras signalling cascade, p120RasGAP global knockout (KO) mice exhibit major vascular defects that result in embryonic lethality[16]. Owing to its ability to interact with a wide array of SH2-binding proteins, p120RasGAP has, in recent years, also been shown to control the return of endocytosed integrin receptors back to the plasma membrane via interactions with the cytoplasmic tail of the integrin α-subunit. In MDA-MB-231 breast cancer cells, for example, p120RasGAP was demonstrated to competitively displace Rab21 from endocytosed β1 integrin in early endosomes, expediting its recycling via Rab11[+] endosomes[17]. Indeed, integrin traffic is critically dependent upon the ability of Rab proteins to dynamically switch between GTP- and GDP-bound states: mutant GDP-locked or GTP-locked Rab21, Rab5, Rab4 and Rab11 all blocking intracellular transit[4,18]. Here, we show that p120RasGAP promotes ligand-bound α5 integrin subunit recycling via the Rab11 compartment in microvascular ECs. α5 integrin is first endocytosed to early endosomes in complex association with both NRP1 and NRP2 co-receptors, where p120RasGAP facilitates focal adhesion kinase (FAK) phosphorylation-dependent transit to Rab11[+] recycling endosomes. siRNA silencing of both NRPs, p120RasGAP, Rab11 or inhibition of FAK phosphorylation results in the intracellular accumulation of α5 integrin. Utilising comparative proteomics analyses, we identify an upregulated escape mechanism to divert α5 integrin away from lysosomal degradation, preserving its total and surface-level expression. In addition, we isolate both NRP1-dependent and NRP2-dependent α5 integrin interactomes in ECs and where they intersect during intracellular traffic. Using inducible endothelial-specific knockout mouse models, we validate these findings by demonstrating a functional loss of FN in the postnatal retina upon co-deletion of both NRP1 and NRP2, which severely impairs polarised sprouting.

## Results

### A proteome-scale map of the α5 integrin interactome in unstimulated microvascular ECs

Both NRP1 and NRP2 have been independently identified as key regulators of α5 integrin trafficking and fibronectin (FN) fibrillogenesis in ECs[8–10]. However, the redundancies that exist between NRP1- and NRP2-directed transport of α5 integrin, or indeed their ability to crosstalk during this process, remains unclear. Furthermore, the interacting complexes that participate in NRP-dependent α5 integrin traffic are poorly annotated.

To begin addressing this question, we employed label-free quantitative data-independent acquisition mass spectrometry (LFQ-DIA-MS) to acquire an unbiased insight into the molecular binding partners of α5 integrin. To this aim, proteomic analysis of WT mouse-microvascular EC lysates immunoprecipitated with α5 integrin from a total of three independent experiments was performed, alongside Tat-Cre-deleted ECs lacking expression of α5 integrin (α5.EC[KO]) to isolate non-specific partner binding. The Spectronaut label-free quantification (LFQ) algorithm was subsequently used for protein quantification with a p value cutoff of <0.01. A total of 3043 proteins were identified as putative interactors of α5 integrin, including fibronectin (Fn1), β1 integrin subunit (Itgb1), NRP1 (Nrp1) and NRP2 (Nrp2), indicating that our immunoprecipitation strategy had been successful (Fig. 1a–c). Pathway enrichment analysis of these 3043 proteins revealed a high proportion to be involved in receptor endocytosis and trafficking, including 18 RabGTPase proteins, 18 myosin motor proteins, clathrin heavy chain-1 (Clt) (and its associated AP2 adaptor complex), caveolae-associated protein-1 (Cavn1), early endosome antigen-1 (EEA1) and Gipc1 (Fig. 1d and Suppl. Fig. 1A–D).

Given the previously annotated roles of NRP1 and NRP2 during α5 integrin traffic[8–10], we proceeded to examine their interaction more closely. To this end, we confirmed a robust biochemical interaction by co-immunoprecipitation and Western blotting. The stoichiometry of this interaction was also greatly increased following a 10-min pre-treatment with the recycling inhibitor primaquine (PMQ), which has been demonstrated to preserve endosomal-integrin contacts in the cell[8], suggesting that NRP—α5 complexes exist more stably within endosomal punctae rather than at the plasma membrane (Fig. 1e–g). Subsequent immunofluorescence confocal microscopy analysis to determine the subcellular localisation of both NRPs and α5 integrin revealed all three colocalise within endosomal structures, and albeit weakly, within proximity of fibrillar adhesions (Fig. 1h), confirming our immunoprecipitation studies.

Next, we visualised the spatial segregation between endogenous NRP1 and NRP2 in endocytic compartments where α5 integrin has been reported to reside in, including EEA1[+]/Rab21[+] early endosomes[4,17], Rab4[+]/Rab11[+] recycling endosomes[9,19] and Rab7[+] late endosomes[19]. Importantly, confocal microscopy in fixed ECs revealed both NRPs to initially colocalise with α5 integrin in clathrin budding coats positive for both clathrin heavy chain-1 and dynamin-2 (Fig. 1i–k). Furthermore, NRP1 and NRP2 were found to share a surprisingly high colocalisation fidelity with each other in EEA1[+], Rab21[+], Rab11[+] and Rab7[+] (but not in Rab4[+]) punctae (Fig. 1l–n).

To corroborate our MS and confocal studies, we silenced expression of NRP1 and NRP2 individually (siNRP1, siNRP2) or simultaneously (siNRP1/2) by RNA interference (RNAi), transfecting mouse-microvascular ECs with either non-targeting (Ctrl) or NRP-specific siRNA oligonucleotides (Fig. 1o and Suppl. Fig. 1E). After confirming the absence of any gene expression changes following siRNA targeting (Suppl. Fig. 1F), we proceeded to examine α5 integrin colocalisation with endogenous Rab11. Depletion of either NRP1 or NRP2 significantly reduced the number of α5 integrin[+] Rab11 endosomes (Fig. 1p, q), confirming previous studies, and indicating a reciprocal contribution in the Rab11[+] long-loop recycling pathway.

### Assembly of α5 integrin[+] adhesions and EDA-FN fibrillogenesis is impaired following NRP1/2 co-depletion in ECs

In ECs, both α5 and αV integrins are known to bind the extracellular matrix (ECM) ligand fibronectin (FN)[1,20–24]. To begin investigating the contributions of NRP1 and NRP2 during α5 integrin-dependent adhesion to FN, a process that relies on receptor trafficking[7], we seeded Ctrl and siRNA-depleted ECs onto FN and assessed their ability to adhere over 3 h. siRNA-mediated depletion of either NRP1 or NRP2 significantly impaired adhesion to FN over this period as previously described[8,10]. Co-depleting both NRP1 and NRP2, however, was found to rescue EC adhesion at 90 min and significantly ameliorate EC adhesion at 180 min (Fig. 2a). Confocal microscopy imaging to visualise endogenous α5 and αV integrin[+] adhesions, together with the focal adhesion (FA) adaptor protein paxillin in ECs fixed at 180 min ± VEGF stimulation subsequently revealed significant reductions in paxillin, α5 and αV integrin[+] adhesion area and number in siNRP1 and siNRP2 ECs, substantiating results by Ruhrberg et al., Alghamdi et al. and Benwell et al. and

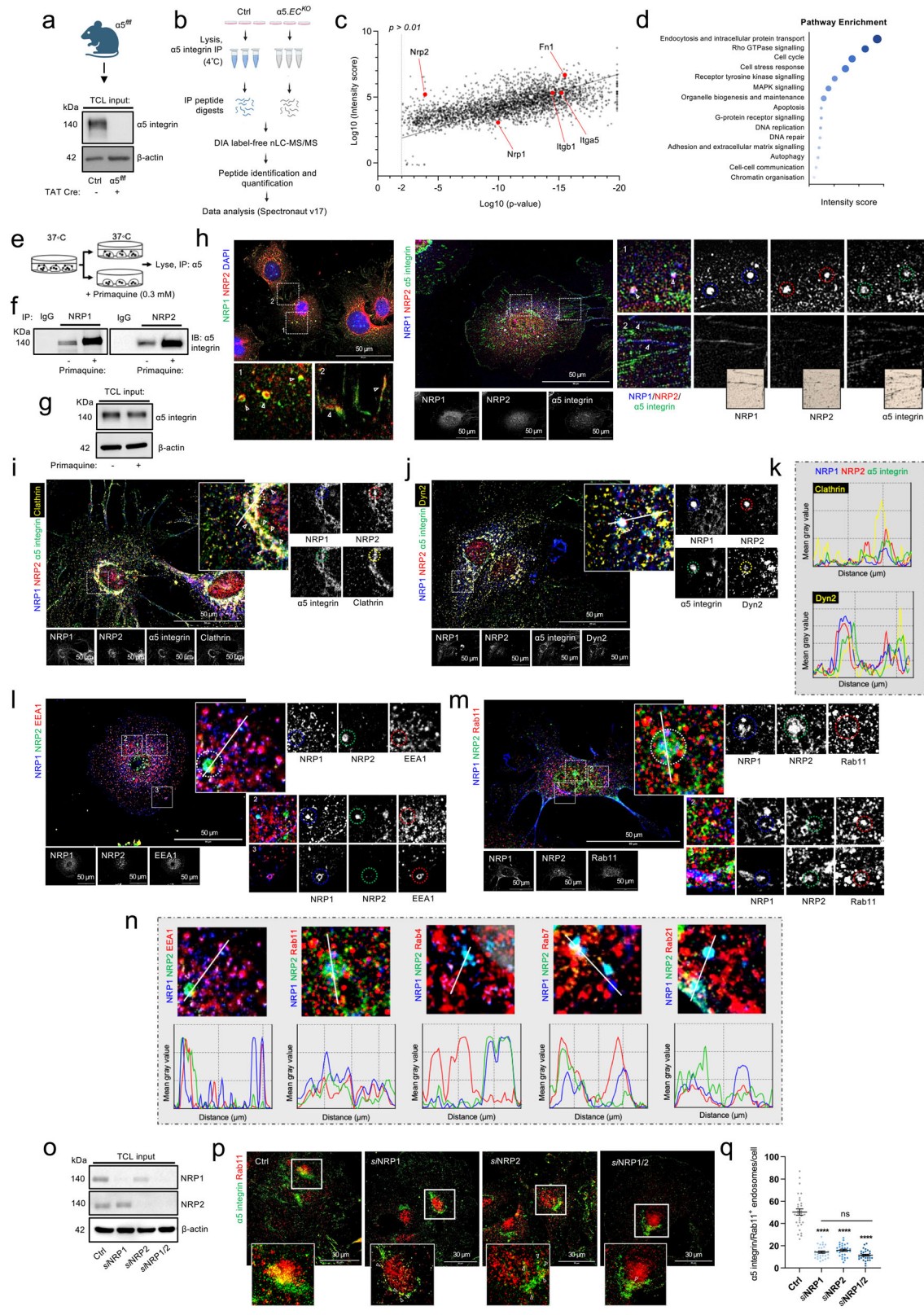

our adhesion studies[8–10]. In comparison, *si*NRP1/2 ECs assembled large punctate paxillin and αV integrin+ adhesions, but very few α5 integrin+ adhesions (Fig. 2b–d and Suppl. Fig. 2A). High resolution XZ-ratio confocal imaging subsequently confirmed paxillin+ adhesions to be largely absent of α5 integrin in ECs depleted for both NRP1 and NRP2 (Fig. 2e). Photo-activation studies employing a tdTomato-expressing α5 integrin construct

validated these findings (Fig. 2f). Surprisingly, no gross changes in total endogenous expression of either integrin or paxillin between our siRNA-depleted groups were observed (Fig. 2g and Suppl. Fig. 2B).

Hypothesising that αV integrin likely compensates for the loss of α5 integrin from adhesions in *si*NRP1/2 ECs, as alluded to by Van der Flier et al.[23], we next employed the αVβ3-specific Cilengitide mimetic

**Fig. 1 | A proteome-scale map of the α5 integrin interactome in unstimulated microvascular ECs. a** Representative Western blot image showing confirmation of TAT-Cre-mediated deletion of α5 integrin from immortalised mouse-microvascular ECs isolated from α5$^{fl/fl}$ mice. **b** Schematic representation of DIA nLC-MS/MS workflow from Ctrl and α5.EC$^{KO}$ EC lysates from three independent experiments. **c** Scatter plot showing the α5 integrin interactome in mouse-microvascular ECs ($p < 0.01$ cutoff). Highlighted protein hits are shown in dark blue. **d** Pathway enrichment analysis showing the top 15 enriched pathways associated with the α5 integrin interactome. **e** Experimental assay schematic: immunoprecipitation from EC lysates of NRP1 and NRP2 ± pre-treatment with primaquine (0.3 mM), followed by SDS-PAGE and Western blotting with anti-α5 integrin antibody. **f** Western blotting showing NRP interactions with α5 integrin ± pre-treatment with primaquine. **g** Total cell lysates showing α5 integrin expression in ECs ± pre-treatment with primaquine. **h** (Left panels): Representative confocal microscopy showing colocalisation between

NRP1 and NRP2 within endosomal and adhesive structures. (Right panels): Representative confocal microscopy showing positive colocalisation between NRP1, NRP2 and α5 integrin. **i–k** Representative confocal microscopy showing positive colocalisation between NRP1, NRP2 α5 integrin and either clathrin heavy chain-1 (**i**) or dynamin-2 (**j**). **l**, **m** Representative confocal microscopy showing positive colocalisation between NRP1, NRP2 and either EEA1 (**l**) or Rab11 (**m**). **n** Mean fluorescence intensity maps showing colocalisation between NRP1, NRP2 and EEA1, Rab11, Rab4, Rab7 or Rab21 respectively. **o** Total cell lysate input showing siRNA depletion of NRP1 and NRP2 by Western blotting. **p** Representative confocal microscopy images showing colocalisation between α5 integrin and Rab11 in Ctrl and siRNA-depleted ECs. **q** Quantification of α5 integrin$^{+}$ Rab11 endosomes/cell in Ctrl and siRNA-depleted ECs, ($n = 30$ cells from three independent biological replicates), one-way ANOVA + post hoc multiple comparisons tests, ****$p < 0.0001$, ns non-significant. **a**, **b**, **e** created with BioRender.com.

EMD66203[25] to inhibit αVβ3-specific adhesion to FN. Ctrl siRNA ECs treated with EMD66203 exhibited partially reduced adhesion to FN at 180 min as expected and assembled smaller paxillin$^{+}$ adhesions. Simultaneous inhibition of αVβ3-specific adhesion via EMD662023, alongside co-depletion of NRP1 and NRP2, resulted in a near total loss of adhesion to FN; however, ECs exhibiting severe defects in cell spreading and adhesion assembly (Fig. 2h–k). Taken together, we infer that additional loss of αVβ3-specific adhesion to FN sensitises ECs to the loss of α5 integrin-dependent adhesion exhibited following depletion of both NRP1 and NRP2.

Since siNRP1/2 ECs failed to assemble α5 integrin$^{+}$ adhesions, but displayed no reduced total expression, we proceeded to determine the subcellular localisation of α5 integrin following NRP depletion by immunofluorescence confocal microscopy. Compared to Ctrl ECs, where α5 integrin is distributed to both surface FAs and to perinuclear punctae, we observed α5 integrin to preferentially accumulate in EEA1$^{+}$ early endosomes upon NRP co-depletion, particularly following VEGF stimulation (Fig. 2l–m).

As stress-fibre-associated FAs become increasingly influenced by actomyosin tension, engaged α5 integrin is translocated centripetally with the actin-binding protein tensin-1 towards the cell body. As a consequence, following 24–48 h adhesion to FN, ECs exhibit a high density of hyper-extended fibrillar adhesions enriched for both α5 integrin and tensin-1[26–28]. The actomyosin tension accrued by α5 integrin translocation into fibrillar structures is also fundamental for FN dimer unfolding and subsequent FN fibrillogenesis[3,29]. Since α5 integrin failed to translocate to adhesion punctae by 180 min, but instead preferentially distributed to intracellular sorting endosomes in siNRP1/2 ECs, we asked whether fibrillar adhesion assembly would also be affected. By 24 h, both siNRP1 and siNRP2 ECs displayed significantly smaller fibrillar adhesions than Ctrl ECs, whilst ECs co-depleted for both NRP1 and NRP2 failed to assemble α5 integrin$^{+}$ or tensin-1$^{+}$ fibrillar adhesions entirely (Fig. 3a, b). This was not found to result from changes in tensin-1 expression (Suppl. Fig. 3A, B). Subsequent immunofluorescent visualisation of endogenous extra domain-A (EDA)- containing cellular FN (EDA-FN) (a spliced isoform of endothelial FN used to detect cell-secreted FN specifically[3,29]) revealed a concomitant loss of fibrillar FN in siNRP1/2 ECs compared to Ctrl ECs or ECs individually depleted for either NRP1 or NRP2 as described previously[13] (Fig. 3c, d). Furthermore, the compensatory actions of αVβ3 integrin observed at 180 minutes were insufficient to rescue cell adhesion (Fig. 3e).

Rather than localise at or in close proximity to fibrillar structures, we observed both α5 integrin and EDA-FN to preferentially accumulate in EEA1$^{+}$ (Fig. 3f, g) and Rab7$^{+}$ late endosomal punctae (Fig. 3h, i). These data indicate that in the absence of NRP1 and NRP2, ligand-bound α5 integrin accumulates intracellularly after failing to recycle via Rab11 from early endosomes.

**p120RasGAP regulates the plasticity of fibrillar adhesions and EDA-FN fibrillogenesis by mediating Rab11-dependent α5 integrin recycling in ECs**

The ubiquitously expressed endosomal GTPase-activating protein p120RasGAP has previously been shown in breast cancer cells to control the return of endocytosed integrin to the plasma membrane. Following Rab21-

mediated integrin endocytosis to the early endosome, p120RasGAP, via a direct mutually exclusive interaction with the cytoplasmic tail of the integrin α-subunit, competes for integrin binding and displaces Rab21. In the absence of p120RasGAP expression, β1 integrin was found to accumulate in EEA1$^{+}$ endosomes, failing its exit to Rab11$^{+}$ recycling endosomes[17]. It remains unclear however, whether p120RasGAP-dependent integrin recycling remains conserved in ECs.

Whilst our MS failed to detect p120RasGAP as directly associating with α5 integrin, a number of other GTPase interactors previously alluded to regulate endosomal transport were detected (RasGTPase-activating-like protein-1 (Iqgap1), RasGTPase-activating protein-binding protein-1 and -2 (G3bp1, G3bp2), Ras-interacting protein-1 (Rasip1), and RasGTPase-activating protein-like-2 (Rasal2)[30–34]. STRING analysis was subsequently employed to generate an interaction network of these proteins, which revealed all to have been previously reported as interactors of p120RasGAP (Suppl. Fig. 4A). We postulate, therefore, that any interaction made with α5 integrin, as signposted by Mai et al.[17] exists weakly, and or very transiently via these adaptors.

To answer whether p120RasGAP promotes endosomal shuttling of α5 integrin in ECs, we first visualised the subcellular localisation of endogenous p120RasGAP in mouse-microvascular ECs by immunofluorescence confocal microscopy, and observed a strong colocalisation with α5 integrin, EEA1$^{+}$ endosomes and Rab21$^{+}$ endosomes (Fig. 4a, b). Next, by means of surface biotinylation followed by biochemical immunoprecipitation (Suppl. Fig. 4B), we assessed the impact of p120RasGAP siRNA-mediated depletion on total α5 integrin recycling. p120RasGAP silencing (Suppl. Fig. 4C) was found to significantly impair the rate at which α5 integrin was recycled back to the plasma membrane compared to Ctrl ECs (Fig. 4c, d and Suppl. Fig. 4D), concomitant with a significant reduction in the localisation of α5 integrin to Rab11$^{+}$ punctae, recapitulating a co-depletion of NRP1 and NRP2 (Fig. 4e, f).

To explore a functional association between Rab21 and p120RasGAP during α5 integrin trafficking, we then sought to impair the endocytosis of α5 integrin from fibrillar adhesions, alongside simultaneous loss of recycling to the membrane. First, we silenced Rab21 expression (to inhibit internalisation) (Suppl. Fig. 4E) and confirmed the defective translocation of α5 integrin from fibrillar adhesions to EEA1$^{+}$ early endosomes (Fig. 4g, h). We subsequently co-silenced p120RasGAP (to inhibit recycling) and examined the localisation of α5 integrin and EDA-FN (Fig. 4i). Compared to Ctrl ECs, where α5 integrin colocalised with tensin-1$^{+}$ characteristic hyperextended structures, α5 integrin was lost completely from fibrillar adhesions in ECs depleted for p120RasGAP and accumulated with EDA-FN in EEA1$^{+}$ early endosomes, recapitulating the loss of NRP1 and NRP2. Importantly, simultaneous depletion of both Rab21 and p120RasGAP preserved α5 integrin localisation to fibrillar adhesions and restored EDA-FN polymerisation into a fibrillar network (Fig. 4j–o). Taken together, these data indicate that p120RasGAP promotes Rab11$^{+}$ α5 integrin recycling from early endosomes, and that the plasticity of fibrillar adhesions and EDA-fibrillogenesis depends on a functional association between Rab21 and p120RasGAP to mediate α5 integrin endocytosis and recycling respectively.

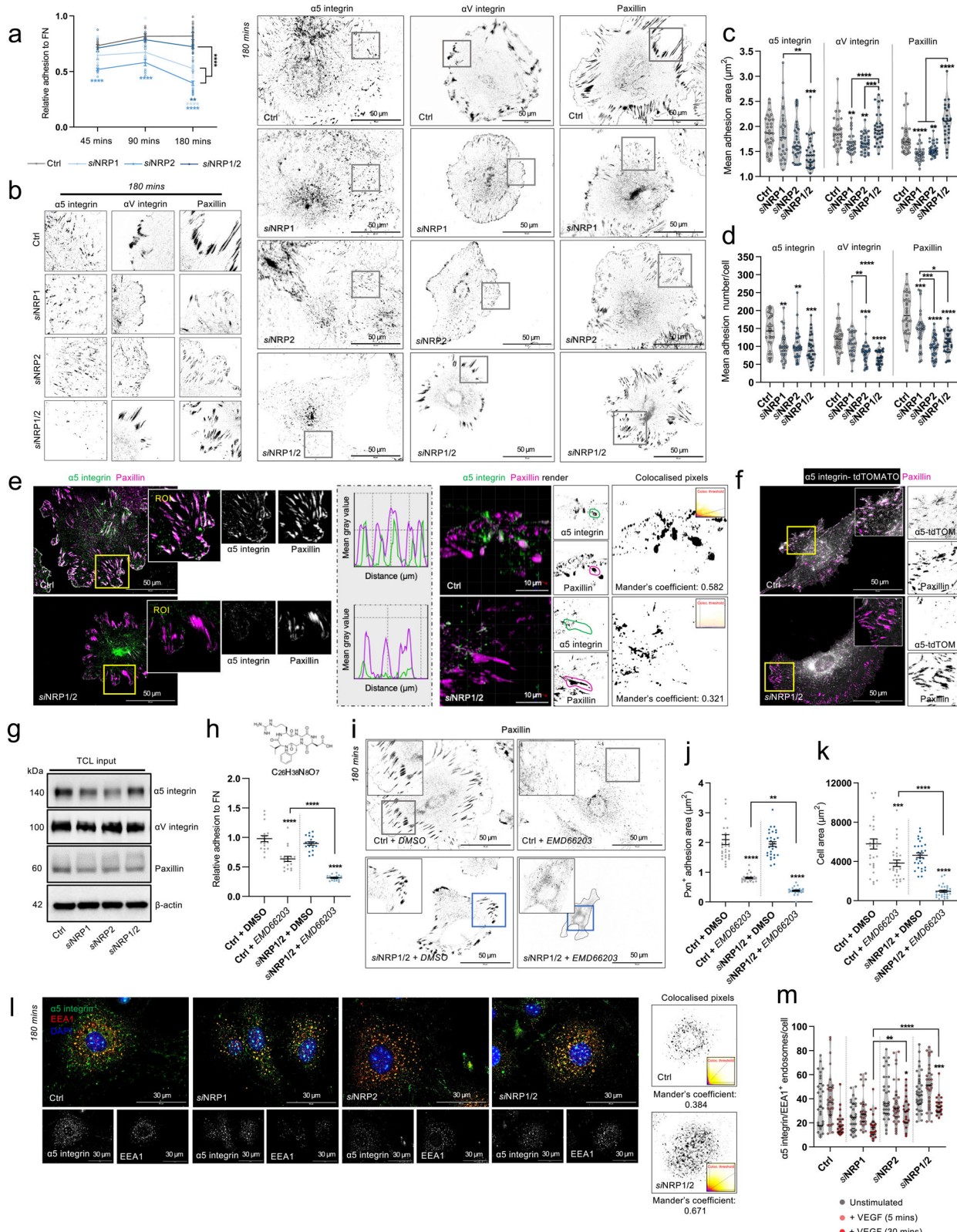

## FAK phosphorylation mediates p120RasGAP interactions with NRP - α5 integrin complexes

Concomitant to its interaction with integrin α-subunit tails, p120RasGAP has also been reported to interact with phosphorylated focal adhesion kinase (*p*FAK), which is brought into close proximity following integrin complex endocytosis to the early endosome[35].

With this in mind, we next determined whether NRP complexes interact with p120RasGAP in ECs during their transit to Rab11+ recycling endosomes, and whether this interaction is indeed dependent upon FAK activity. Lysates from ECs oligofected with either non-targeting Ctrl or NRP-specific siRNAs were immunoprecipitated with antibodies recognising either NRP1 or NRP2, before Western blotting for p120RasGAP. Both

**Fig. 2 | Assembly of α5 integrin⁺ adhesions is impaired following NRP1/2 co-depletion in ECs. a** Relative adhesion to FN by NRP1, NRP2 and NRP1/2 siRNA-treated ECs, ($N = 3$ independent experiments), one-way ANOVA + post hoc multiple comparisons tests, $**p < 0.01$, $****p < 0.0001$. **b** Representative confocal microscopy images showing α5 integrin, αV integrin and paxillin⁺ adhesions in Ctrl, *si*NRP1, *si*NRP2 and *si*NRP1/2 ECs following 180 mins adhesion to FN. **c, d** Analysis of α5 integrin, αV integrin and paxillin⁺ adhesion size ($\mu M^2$) (**c**) and number (**d**) in unstimulated Ctrl, *si*NRP1, *si*NRP2 and *si*NRP1/2 ECs, ($n = 30$ cells from three independent biological replicates), one-way ANOVA + post hoc multiple comparisons tests, $*p < 0.05$, $**p < 0.01$, $***p < 0.001$ and $****p < 0.0001$.
**e** Representative confocal microscopy images showing colocalisation between α5 integrin and paxillin in Ctrl and *si*NRP1/2 ECs in unstimulated conditions. Images are shown alongside mean fluorescence intensity maps. **f** Representative confocal microscopy images showing colocalisation between tdTOMATO expressing α5 integrin and paxillin in Ctrl and *si*NRP1/2 ECs. **g** Total cell lysate input showing

expression levels of α5 integrin, αV integrin and paxillin by Western blotting. **h** Top panel: chemical structure of EMD66203 compound. Bottom panel: relative adhesion to FN by Ctrl and *si*NRP1/2 ECs ± 50 μM EMD66203, ($N = 2$ independent experiments), one-way ANOVA + post hoc multiple comparisons tests, $****p < 0.0001$. **i** Representative confocal microscopy images showing paxillin⁺ adhesions presented by Ctrl and *si*NRP1/2 ECs ± 50 μM EMD66203 following 180 min adhesion to FN. **J, k** Quantification of paxillin⁺ adhesion size (μM) (**j**) and cell area ($\mu M^2$) (**k**) of Ctrl and *si*NRP1/2 ECs ± 50 μM EMD66203, ($n = 30$ cells from three independent biological replicates), one-way ANOVA + post hoc multiple comparisons tests, $**p < 0.01$, $***p < 0.001$, $****p < 0.0001$. **l** Representative confocal microscopy images showing α5 integrin⁺ EEA1 early endosomes in Ctrl, *si*NRP1, *si*NRP2 and *si*NRP1/2 ECs following 180 min adhesion to FN. **m** Quantification of α5 integrin⁺ EEA1 endosomes/cell ± 5 or 30 mins VEGF₁₆₅ stimulation, ($n = 30$ cells from three independent biological replicates), one-way ANOVA + post hoc multiple comparisons tests, $**p < 0.01$, $***p < 0.001$, $****p < 0.0001$.

---

NRP1 and NRP2 co-immunoprecipitated with p120RasGAP, however the biochemical interaction between NRP2 and p120RasGAP was significantly increased following NRP1 silencing (Fig. 5a–d), indicating a degree of compensation during complex assembly. To establish whether these interactions were mediated by FAK phosphorylation, we employed the well-characterised FAK-specific inhibitor PF562271, with which we initially confirmed a potent inhibition of FAK activation at both Tyr³⁹⁷ and Tyr⁴⁰⁷ phosphorylation sites at a concentration of >1 μM as previously described[36–38] (Fig. 5e). PF562271 pre-treatment prior to lysis was found to ablate not only interactions with NRP1, but also α5 integrin and total FAK (Fig. 5f, g), validating prior works describing a role for *p*FAK in mediating complex interactions with p120RasGAP[35]. Whilst NRP1–p120RasGAP complex association was found to be *p*FAK-dependent, PF562271 pre-treatment had no effect on the association between p120RasGAP and NRP2 however. Only when NRP1 expression was depleted, where previously we observed a significantly greater interaction, did their affinity become dependent upon FAK activity (Fig. 5h). These data suggest that NRP1 and NRP2 competitively bind p120RasGAP during their transport of α5 integrin–FAK complexes, and that only in the absence or downregulated expression of NRP1 does NRP2 associate.

To corroborate our findings, we proceeded to examine the effect of PF562271 pre-treatment on α5 integrin subcellular localisation in Ctrl ECs. PF562271-elicited inhibition of FAK phosphorylation recapitulated the loss of α5 integrin colocalisation with Rab11⁺ endosomes and the concomitant accumulation in EEA1⁺ early endosomes observed following NRP co-depletion or silencing of p120RasGAP. Furthermore, whilst PF562271 incubation had no effect on α5 integrin - Rab4 colocalisation, significantly more α5 integrin was observed to reside in Rab7⁺ compartment punctae, suggesting that following α5 integrin accumulation in early endosomes, it is diverted for lysosomal processing (Fig. 5i, j).

### Retrograde transport via the Dynein–Dynactin complex saves α5 integrin from lysosomal degradation

Despite α5 integrin accumulating in EEA1⁺ early endosomes and Rab7⁺ late endosomes, both surface and total α5 integrin expression were observed to remain intact in *si*NRP1/2 and *si*p120RasGAP ECs (Fig. 6a–c). This led us to consider whether, in the absence of recycling via the Rab11 compartment, α5 integrin utilises an alternative trafficking route to avoid lysosomal degradation. In an attempt to elucidate the novel α5 integrin interactors that participate in non-canonical trafficking pathways, we performed comparative proteomic analysis between our α5 integrin interactome dataset acquired from Ctrl EC lysates, and lysates collected from *si*NRP1, *si*NRP2, *si*NRP1/2 and *si*p120RasGAP ECs from a total of three independent experiments in the same manner as described in Fig. 1a–c (Fig. 6d).

First, we established that the interaction between α5 integrin and both NRPs had been reduced to negligible levels, reaffirming the efficiency of our siRNA depletions (Fig. 6e). Next, despite no gross changes in the number of quantified proteins (Fig. 6f), we observed a clear segregation by principal

component analysis (PCA) between our Ctrl datasets and those acquired from NRP1/NRP2 or p120RasGAP depleted lysates, indicating clear alterations to α5 integrin's interactome (Fig. 6g). These changes were subsequently visualised in each of our siRNA-depleted groups with volcano plots, identifying protein candidates with the greatest fold-change differences compared to our Ctrl dataset (with a $p$ value cutoff >0.05) (Fig. 6h).

Prior to evaluating any upregulated trafficking pathways unique to ECs depleted for both NRPs or p120RasGAP, we noted striking differences in the fold-change profiles acquired from our *si*NRP1 and *si*NRP2 EC datasets. In an attempt to isolate NRP1 and NRP2-dependent α5 integrin binding partners, we proceeded to visualise the protein fold-change ratios between *si*NRP2 and *si*NRP1 groups (Fig. 6i). Of these α5 integrin interacting proteins, 68 were associated with α5 integrin at a $\log_2$ fold-change $<-1$ (which we defined as NRP2-dependent interactions), 871 associated at a $\log_2$ fold-change $>1$ (defined NRP1-dependent interactions), and 971 associated at an insignificant $\log_2$ fold-change ($p > 0.05$) (shared interactions) (Fig. 6j). Comparative pathway enrichment analysis to evaluate these NRP-specific interactions revealed both to regulate unique intracellular trafficking pathways utilised by α5 integrin in ECs (46 NRP1-dependent, 16 NRP2-dependent and 122 shared interactions) (Fig. 6k–m). These data support a paradigm whereby NRP1 and NRP2's ability to co-traffic α5 integrin in ECs is context-dependent. Indeed, our findings suggest that both NRP1 and NRP2 independently regulate α5 integrin's association with multiple trafficking proteins, in addition to mediating other interactions cooperatively. For example, and corroborating the work presented in this study, Rab11 and Rab21 were both detected as common denominators to both NRP1 and NRP2, alongside α5 integrin's interactions with Myosin 6, Rab5 and Gipc1 as previously reported[8] (Fig. 6m).

To elucidate non-canonical trafficking pathways upregulated in *si*NRP1/2 and *si*p120RasGAP ECs, we returned, however to our Ctrl comparisons shown in Fig. 6h. Of the 3056 proteins enriched from our *si*NRP1/2 sample, only 50 were identified as interacting with α5 integrin at a positive $\log_2$ fold-change $>1$ compared to our Ctrl group, of which 41 shared a similar positive $\log_2$ fold-change profile with our *si*p120RasGAP group. Of these 41 putative interacting proteins, four exhibited a positive $\log_2$ fold-change $>8$ (fold-change $>3$): Centrosomal protein-63 (Cep63)[39], Dynein light chain-3 (Dynlt3)[40], Interleukin-12 receptor β1 (Il12rβ1)[41] and YTH N6-methyladenosine RNA binding protein C1 (Ythdc1)[42]. Upon inspection of our *si*NRP1 and *si*NRP2 datasets, Cep63, Dynlt3 and Il12rβ1 were all found to interact with α5 integrin at a similarly high magnitude, indicating their reciprocal contribution in the absence of the NRPs or p120RasGAP (Fig. 6n).

Dynlt3 is known to homodimerize to assemble the light chain component of the cytoplasmic dynein motor protein complex, which with its cofactors Dynactin 1–5, promotes retrograde transport from Rab7a⁺ late endosomes along the microtubule cytoskeleton to the trans-Golgi network (TGN)[43–45]. This non-canonical trafficking route has been elucidated to function as a defence mechanism against undesirable protein degradation, diverting integral receptors away from the lysosome and redirecting them

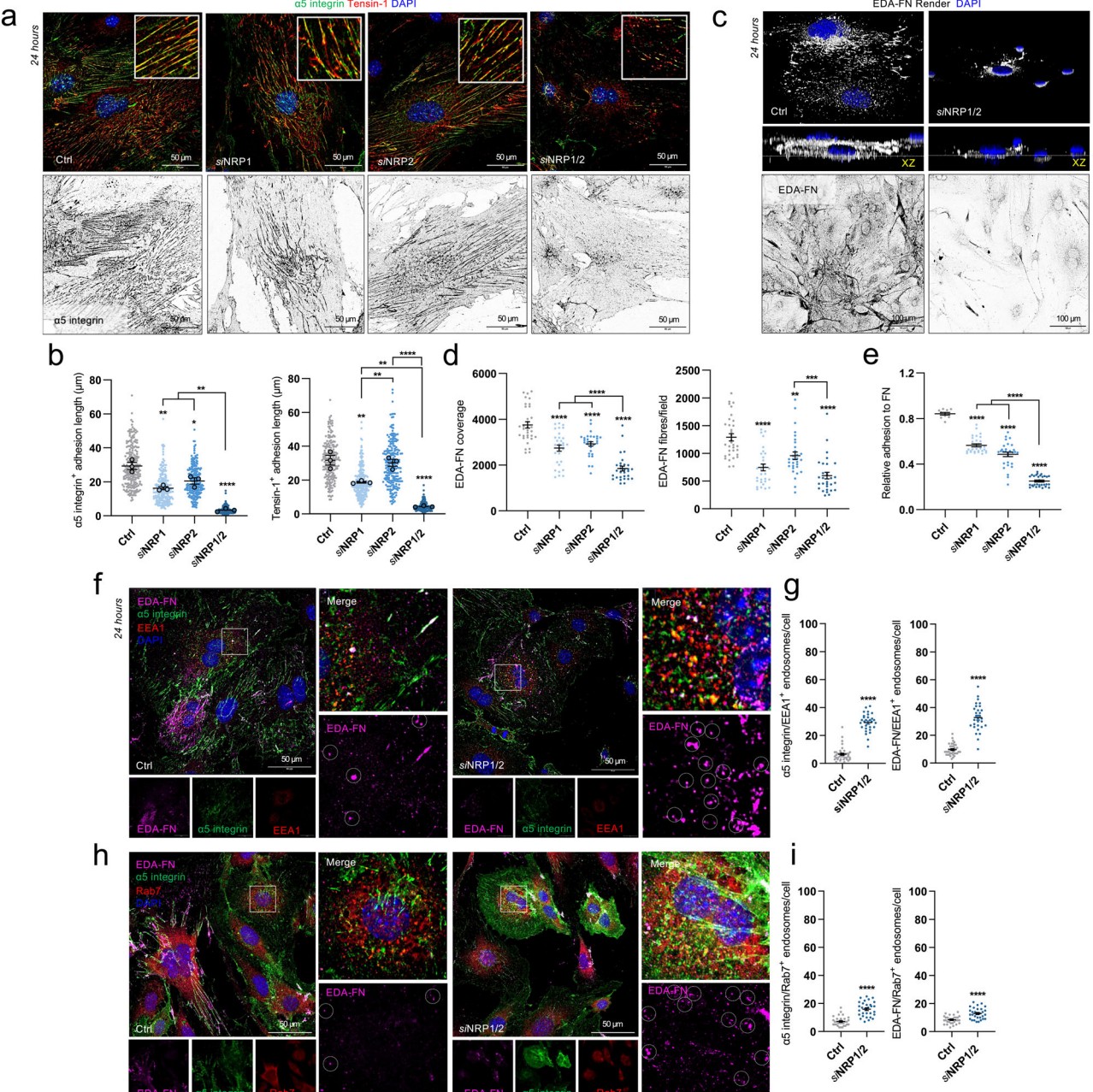

**Fig. 3 | Development of α5 integrin⁺ fibrillar adhesions and subsequent EDA-FN fibrillogenesis is lost in *si*NRP1/2 ECs. a** Representative confocal microscopy images showing α5 integrin⁺ and tensin-1⁺ fibrillar adhesions in Ctrl, *si*NRP1, *si*NRP2 and *si*NRP1/2 ECs following 24 h adhesion to FN. **b** Quantification of mean α5 integrin⁺ and tensin-1⁺ adhesion length (μm), (*n* = 30 cells from three independent biological replicates), one-way ANOVA + post hoc multiple comparisons tests, **p* < 0.05, ***p* < 0.01, *****p* < 0.0001. **c** Representative confocal microscopy images showing endogenous EDA-FN in Ctrl, *si*NRP1, *si*NRP2 and *si*NRP1/2 ECs following 24 h adhesion to FN. Middle panels show rendered EDA-FN in Ctrl and *si*NRP1/2 ECs with XZ-ratio. **d** Quantification of mean EDA-FN intensity and mean EDA-FN fibre density from confluent ECs, (*n* = 30 ROIs from three independent biological replicates), one-way ANOVA + post hoc multiple comparisons tests,

***p* < 0.01, *****p* < 0.0001. **e** Relative adhesion to FN by NRP1, NRP2 and NRP1/2 siRNA-treated ECs at 24 h, (*N* = 3 independent experiments), one-way ANOVA + post hoc multiple comparisons tests, *****p* < 0.0001. **f** Representative confocal microscopy images showing colocalisation between EDA-FN, α5 integrin and EEA1 in Ctrl and *si*NRP1/2 ECs following 24 h adhesion to FN. **g** Quantification of α5 integrin⁺ and EDA-FN⁺ EEA1 early endosomes/cell, (*n* = 30 cells from three independent biological replicates), Student's *t*-test, *****p* < 0.0001. **h** Representative confocal microscopy images showing colocalisation between EDA-FN, α5 integrin and Rab7 in Ctrl and *si*NRP1/2 ECs following 24 h adhesion to FN. **i** Quantification of α5 integrin⁺ and EDA-FN⁺ Rab7 late endosomes/cell, (*n* = 30 cells from three independent biological replicates), Student's *t*-test, *****p* < 0.0001.

back to the plasma membrane[46,47]. We hypothesise, therefore, that following the loss of Rab11-dependent recycling, elicited by the depletion of NRP1 and NRP2, or p120RasGAP, α5 integrin's surface (and total) expression is maintained by retrograde transport via the TGN and the dynein–dynactin complex. In support of this hypothesis, α5 integrin's interaction with Dynactin 3, 4 and 5 (Dctn3-5) was significantly elevated in *si*NRP1/2 and

*si*p120RasGAP ECs compared to the Ctrl sample (Fig. 6o). We subsequently validated our proteomics by immunofluorescence imaging, demonstrating that α5 integrin accumulated in dynein⁺ punctae following co-depletion of NRP1 and NRP2, or by silencing p120RasGAP expression (Fig. 6p, q). In sum, proteomics analyses has revealed that α5 integrin requires NRP1 and NRP2 to direct polarised transport through its canonical recycling pathway,

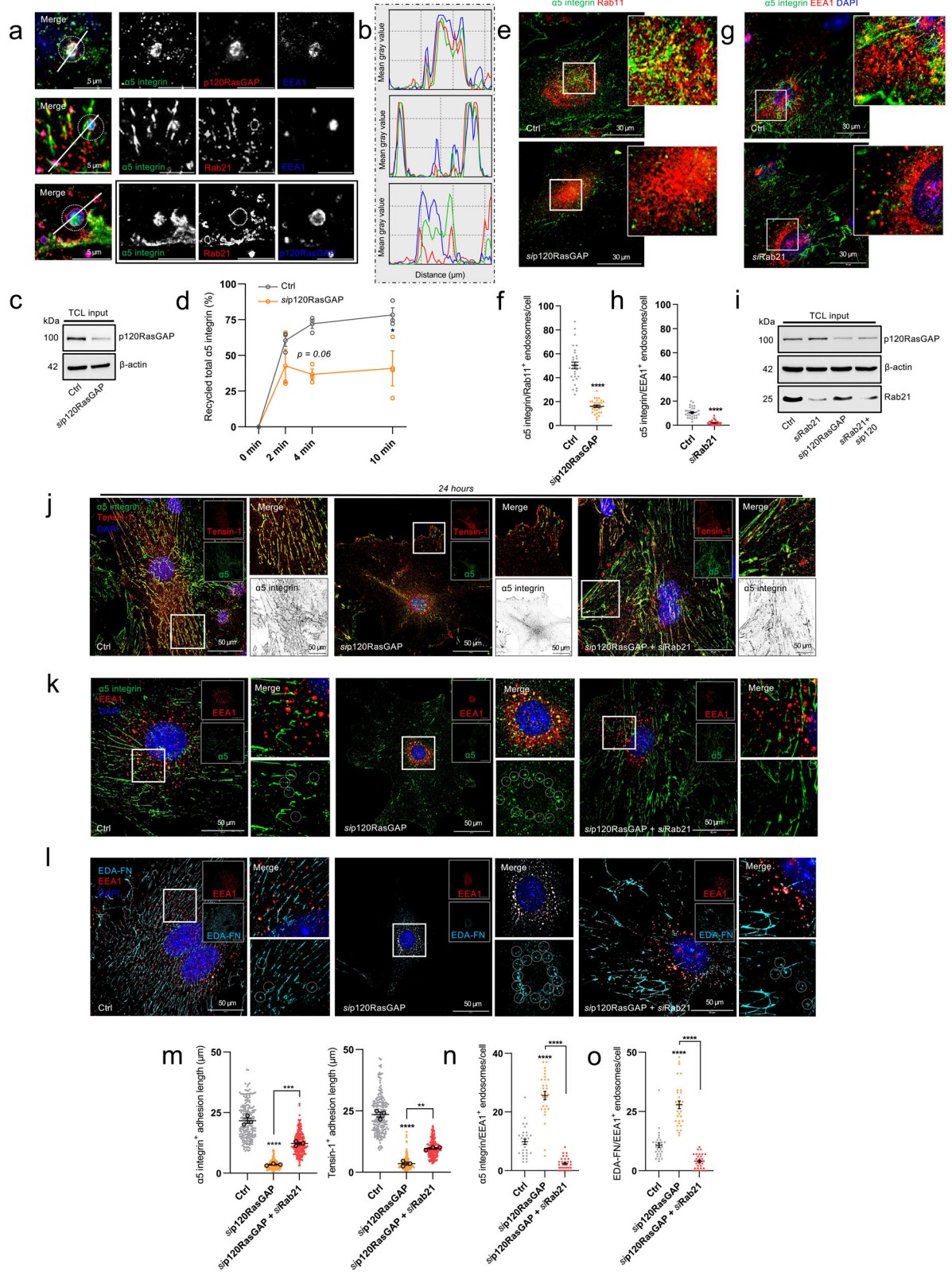

and in their absence, ECs adopt a rescue mechanism to divert α5 integrin away from lysosomal degradation. In addition, we have identified a plethora of previously unknown, NRP-specific α5 integrin interactions. These putative associations encompass not only endosomal trafficking, as discussed here, but also cell cycle regulation, organelle biogenesis, cell-stress responses, and a range of canonical signalling pathways.

## Endothelial NRPs and p120RasGAP promote cell polarity during directional cell migration

The SH2 domain-dependent pFAK association with p120RasGAP has been shown to promote cell polarisation in mouse embryonic fibroblasts, where silencing of p120RasGAP or inhibition of FAK phosphorylation results in disruptions to Golgi body orientation and subsequent impaired directional

**Fig. 4 | p120RasGAP regulates the plasticity of fibrillar adhesions and EDA-FN fibrillogenesis by mediating Rab11-dependent α5 integrin recycling in ECs.**
**a** Representative confocal microscopy images showing colocalisation between α5 integrin, p120RasGAP and EEA1 (top panels), α5 integrin, Rab21 and EEA1 (middle panels), and α5 integrin, Rab21 and p120RasGAP (bottom panels). **b** Mean fluorescence intensity maps showing respective colocalisation between α5 integrin, p120RasGAP, EEA1 and Rab21 as shown in (**a**). **c** Total cell lysates shown to confirm siRNA depletion of p120RasGAP. **d** Quantification of total α5 integrin recycling (%), (N = 3 independent experiments), one-way ANOVA + post hoc multiple comparisons tests, *p < 0.05. **e** Representative confocal microscopy images showing colocalisation between α5 integrin and Rab11 in Ctrl and sip120RasGAP ECs. **f** Quantification of α5 integrin+ Rab11 endosomes/cell in Ctrl and siRNA-depleted ECs, (n = 30 cells from three independent biological replicates), Student's t-test, ****p < 0.0001. **g** Representative confocal microscopy images showing colocalisation between α5 integrin and EEA1 in Ctrl and siRab21 ECs. **h** Quantification of α5 integrin+ EEA1 endosomes/cell in Ctrl and siRab21 ECs, (n = 30 cells from three

independent biological replicates), Student's t-test, ****p < 0.0001. **i** Total cell lysates confirming siRNA depletion of p120RasGAP and Rab21. **j** Representative confocal microscopy images showing α5 integrin and tensin-1 localisation to fibrillar adhesions at 24 h adhesion to FN in Ctrl, sip120RasGAP and siRab21 ECs. **k** Representative confocal microscopy images showing α5 integrin colocalisation with EEA1 at 24 h adhesion to FN in Ctrl, sip120RasGAP and siRab21 ECs. **l** Representative confocal microscopy images showing EDA-FN colocalisation with EEA1 at 24 h adhesion to FN in Ctrl, sip120RasGAP and siRab21 ECs. **m** Quantification of α5 integrin and tensin-1+ adhesion length, (n = 30 cells from three independent biological replicates), one-way ANOVA + post hoc multiple comparisons tests, **p < 0.01, ****p < 0.0001. **n** Quantification of α5 integrin+ EEA1 endosomes/cell, (n = 30 cells from three independent biological replicates), one-way ANOVA + post hoc multiple comparisons tests, ****p < 0.0001. **o** Quantification of EDA-FN+ EEA1 endosomes/cell, (n = 30 cells from three independent biological replicates), one-way ANOVA + post hoc multiple comparisons tests, ****p < 0.0001.

migration[14,35]. Long-loop recycling of α5 integrin from fibrillar adhesions and the accompanying exocytosis of fibrillar EDA-FN by Rab11 recycling endosomes has also been demonstrated to establish the apico-basal polarity axis in ECs, which is essential for vascular morphogenesis in vivo[3].

We have demonstrated previously that NRP2 silencing in ECs moderately impairs the response of FAK phosphorylation to VEGF-stimulation[9]. We, therefore, hypothesised that attenuated FAK phosphorylation in ECs depleted for either NRP or p120RasGAP would also impair cell polarisation. To test this, we first assessed FAK phosphorylation at Tyr[397] or Tyr[407] residues, both reported to regulate EC adhesion and migration[9,48], under basal conditions or following VEGF stimulation. When normalised against total FAK expression, silencing of either NRP1, NRP2, or p120RasGAP in confluent ECs stimulated with VEGF for 5 minutes significantly reduced FAK phosphorylation at Tyr[397], and induced a robust reduction in phosphorylation at Tyr[407] (Fig. 7a, b and Suppl. Fig. 5A, B). Expression of both pFAK[Tyr397] and Tyr[407] under basal conditions was also found to be significantly attenuated following co-depletion of both NRP1 and NRP2 (Fig. 7c). To corroborate our biochemical studies, we visualised endogenous pFAK[Tyr407+] by immunofluorescence microscopy in ECs fixed 8 h-post initiation of a scratch-wound to assess their ability to assemble polarised adhesions (Fig. 7d). Whilst Ctrl ECs assembled a higher proportion of leading edge pFAK[Tyr407+] adhesions vs at their trailing edge, indicative of correctly establishing a polarity axis towards the direction of migration, ECs depleted for both NRP1 and NRP2, or p120RasGAP were observed to assemble equal numbers of both leading and trailing edge pFAK[Tyr407+] adhesions (Fig. 7e, f), suggesting an impaired ability to polarise.

To directly quantify the effect of NRP or p120RasGAP silencing on cell polarisation, we subsequently performed scratch-wound assays in confluent Ctrl and siRNA-depleted ECs before analysing Golgi body orientation. Compared to Ctrl ECs, where ~80% of Golgi bodies were orientated towards the direction of the scratch site, silencing of either NRP1 or NRP2 reduced correct Golgi polarisation by approximately half. This was further reduced to ~20 and ~30% following co-depletion of NRP1 and NRP2 or p120Ras-GAP, respectively. Alongside an impaired ability to polarise correctly, siRNA depletion also reduced the number of F-actin+ lamellipodial protrusions directed into the avascular space (Fig. 7g–h). Taken together, we postulate that NRP1 and NRP2 direct α5 integrin long-loop recycling via p120RasGAP in a pFAK-dependent manner to promote polarised EC migration over FN matrices.

Finally, to validate this hypothesis, we assessed the impact of Rab11 silencing on the assembly of α5 integrin and pFAK[Tyr407+] adhesions in ECs. After confirming effective co-depletion of both Rab11a and Rab11b isoforms by Western blotting (Fig. 7i and Suppl. Fig. 5C), we visualised both endogenous α5 integrin and pFAK[Tyr407] in Ctrl and siRab11 ECs fixed at 180 minutes adhesion to FN by immunofluorescence confocal microscopy. Rab11 depletion was found to significantly reduce the number of both α5 integrin and pFAK[Tyr407+] adhesions (Fig. 7j, k) and recapitulated the accumulation of α5 integrin in EEA1+ early endosomes and Rab7+ late

endosomes (Fig. 7l, m and Suppl. Fig. 5D, E) as observed in siNRP1/2 ECs, sip120RasGAP ECs, and ECs pre-treated with PF562271. Finally, siRab11 ECs displayed significantly reduced Golgi body polarisation (Fig. 7n and Suppl. Fig. 5F), further substantiating the plethora of studies describing the role of Rab11 in promoting cell polarisation via polarised transport of endocytosed cargoes[3,49–51].

## Endothelial NRPs are essential for polarised sprouting angiogenesis and FN fibrillogenesis

Polarised FN secretion and directional EC migration are required for establishing a functional vascular network[3]. To determine the role of NRP1 and NRP2 during FN fibrillogenesis in vivo, we utilised NRP1[fl/fl].Pdgfb-iCreER[T2] (NRP1[fl/fl].EC[KO]), NRP2[fl/fl].Pdgfb-iCreER[T2] (NRP2[fl/fl].EC[KO]), and NRP1[fl/fl]NRP2[fl/fl].Pdgfb-iCreER[T2] (NRP1[fl/fl]NRP2[fl/fl].EC[KO]) animals, generated as previously described[13], and compared the effects of an acute endothelial-specific deletion of NRP1, NRP2, or NRP1 and NRP2 respectively during sprouting angiogenesis of the retina at postnatal day 6 (P6). Following P2 - P5 tamoxifen administrations (Fig. 8a), we first confirmed successful depletion of both NRP1 and NRP2 expression from the superficial vascular plexus (Fig. 8b). Loss of NRP1 expression was found to result in major sprouting defects, and a significant reduction in vascular outgrowth from the optic nerve head, as formerly reported[11]. Likewise, though less severe, deletion of endothelial NRP2 resulted in a mild reduction in vascular extension, and significant impediments to sprouting (as previously described)[9]. Unsurprisingly, endothelial knockout of both NRP1 and NRP2 elicited the most deleterious effects on vascular outgrowth, NRP1[fl/fl]NRP2[fl/fl].EC[KO] animals exhibiting severe defects in vascular extension, branching, and total EC density via ERG1/2 staining, in addition to tip cell and filopodia number (Fig. 8c–g and Suppl. Fig. 6A, B). A significant increase in vessel regression assessed by means of enumerating the frequency of collagen IV+ empty sleeves was also observed (Suppl. Fig. 6C, D). These results largely phenocopy those reported by Chen et al., when describing the impairments to retinal angiogenesis following endothelial knockout of p120RasGAP[16]. Despite severe defects to filopodia number in our NRP[fl/fl].EC[KO] mice, no impairments to filopodial length or tortuosity were observed when compared against respective littermate controls (Suppl. Fig. 6E).

Given our earlier findings, we proceeded to consider whether the combined loss of both NRP1 and NRP2 would also result in defects to tip cell polarisation during sprouting angiogenesis. To this end, we analysed front-rear polarity by GM130 labelling and immunofluorescence confocal microscopy in sprouting ECs. Compared to littermate control mice, whose sprouting ECs correctly oriented their Golgi bodies to align with the direction of flow, NRP1[fl/fl]NRP2[fl/fl].EC[KO] animals exhibited a ~60% impaired front-rear Golgi body polarisation (Fig. 8h, i and Suppl. Fig. 6F), validating our in vitro studies.

FN is highly expressed during the development of the retinal vascular plexi and provides an essential scaffold to facilitate polarised migration of

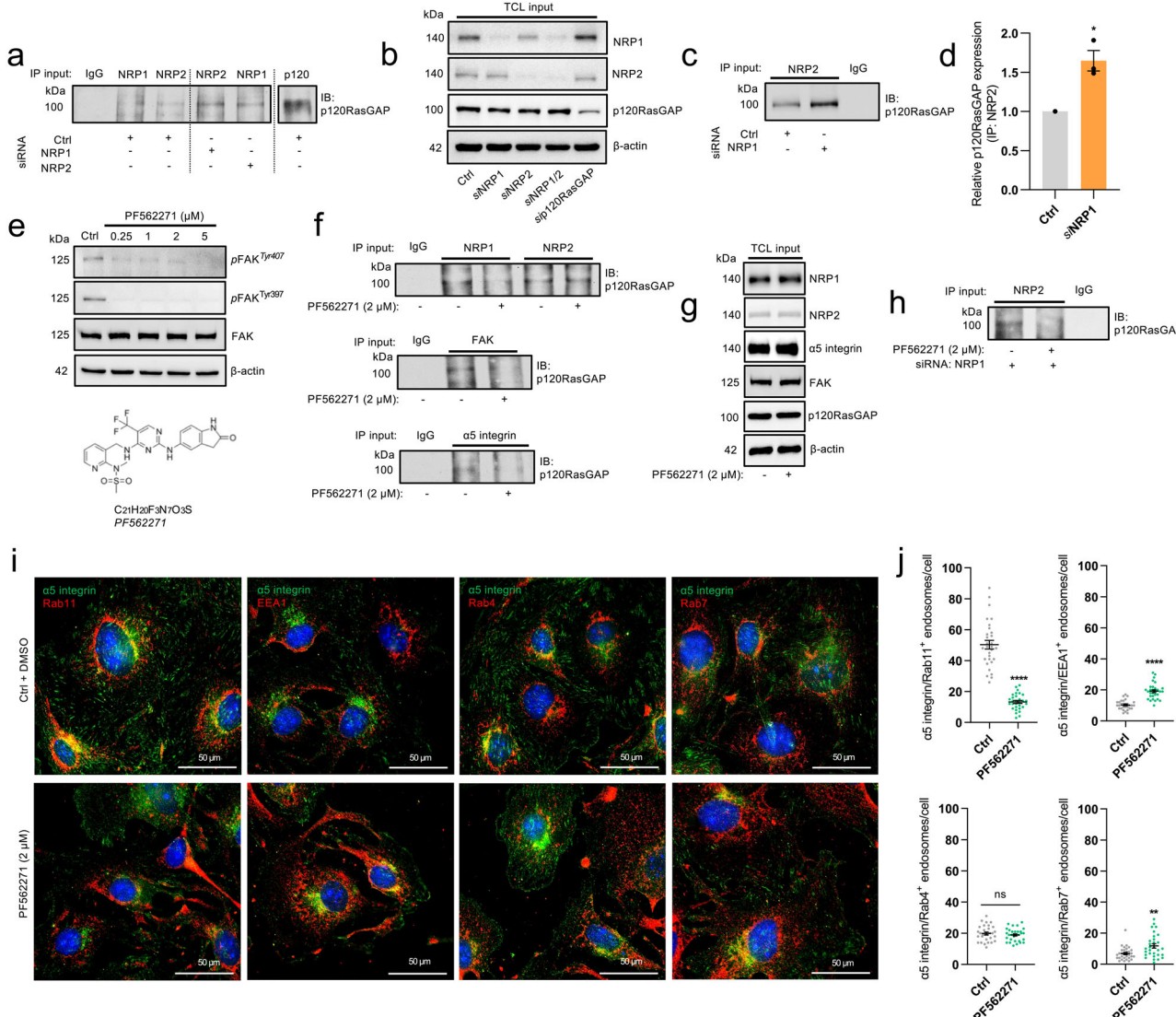

**Fig. 5 | FAK phosphorylation mediates p120RasGAP interactions with NRP - α5 integrin complexes.** **a** Immunoprecipitation from Ctrl and siRNA-depleted EC lysates of NRP1, NRP2 and p120RasGAP followed by SDS-PAGE and Western blotting with anti-p120RasGAP antibody. **b** Total cell lysate input confirming siRNA depletion of NRP1, NRP2 and p120RasGAP by Western blotting. **c** Representative immunoprecipitation from Ctrl and siNRP1-depleted EC lysates of NRP2 followed by SDS-PAGE and Western blotting with anti-p120RasGAP antibody. **d** Quantification of relative p120RasGAP expression in Ctrl and siNRP1 lysates following immunoprecipitation of NRP2, ($N = 3$ independent experiments) one-sample t-test, *$p < 0.05$. **e** Total cell lysate input confirming inhibition of FAK phosphorylation following pre-treatment with 2 µM PF562271. Below shows the chemical structure of PF562271 compound. **f** Immunoprecipitation of NRP1 and NRP2 from EC lysates followed by SDS-PAGE and Western blotting with anti-p120RasGAP antibody showing loss of association following pre-treatment with 2 µM PF562271. **g** Total cell lysate input showing NRP1, NRP2, α5 integrin, FAK and p120RasGAP expression by Western blotting following pre-treatment with 2 µM PF562271. **h** Immunoprecipitation of NRP2 from Ctrl and siNRP1 EC lysates followed by SDS-PAGE and Western blotting with anti-p120RasGAP antibody showing loss of association following pre-treatment with 2 µM PF562271. **i** Representative confocal microscopy images showing α5 integrin colocalisation with Rab11, EEA1, Rab4 or Rab7 in Ctrl ECs ± pre-treatment with PF562271. **j** Quantification of α5 integrin+ Rab11, EEA1, Rab4 or Rab7 endosomes/cell in Ctrl and siRNA-depleted ECs, ±pre-treatment with PF562271 ($n = 30$ cells from three independent biological replicates), Student's t-test, ****$p < 0.0001$.

sprouting ECs towards the retinal periphery[21]. In addition to being assembled ahead of the expanding vascular plexus by retinal astrocytes, EC-derived FN has more recently been shown to be expressed surrounding the vasculature itself[52]. To determine whether endothelial knockout of both NRP1 and NRP2 would impair EC-derived FN fibrillogenesis in vivo, we visualised the expression of EDA-FN in the superficial vascular plexus of P6 mice by immunofluorescence confocal microscopy. In control littermate animals, EDA-FN expression was highest at the vascular front, enriched at the apical ends of sprouting ECs and in the avascular space immediately ahead of the growing vascular bed. Retinal EDA-FN expression was found to be significantly reduced in NRP1[fl/fl]NRP2[fl/fl].EC[KO] mice, however (Fig. 8j, k), an impediment that we postulate would directly impair the ability of new

sprouts to form stable cell-matrix interactions and subsequently polarise to promote directional EC migration (Fig. 9).

## Discussion

In the present study, we show that α5 integrin is endocytosed as a complex with NRP1 and NRP2, before transiting to Rab11+ recycling endosomes via the GTPase-activating protein p120RasGAP. As a consequence of NRP or p120RasGAP silencing, α5 integrin is lost from surface focal and fibrillar adhesions, but rather accumulates with its FN ligand in early and late endosomes. Mechanistically, we show both NRPs biochemically interact with endosomal p120RasGAP, facilitating the displacement of Rab21 to promote traffic to Rab11+ recycling endosomes and subsequent cell

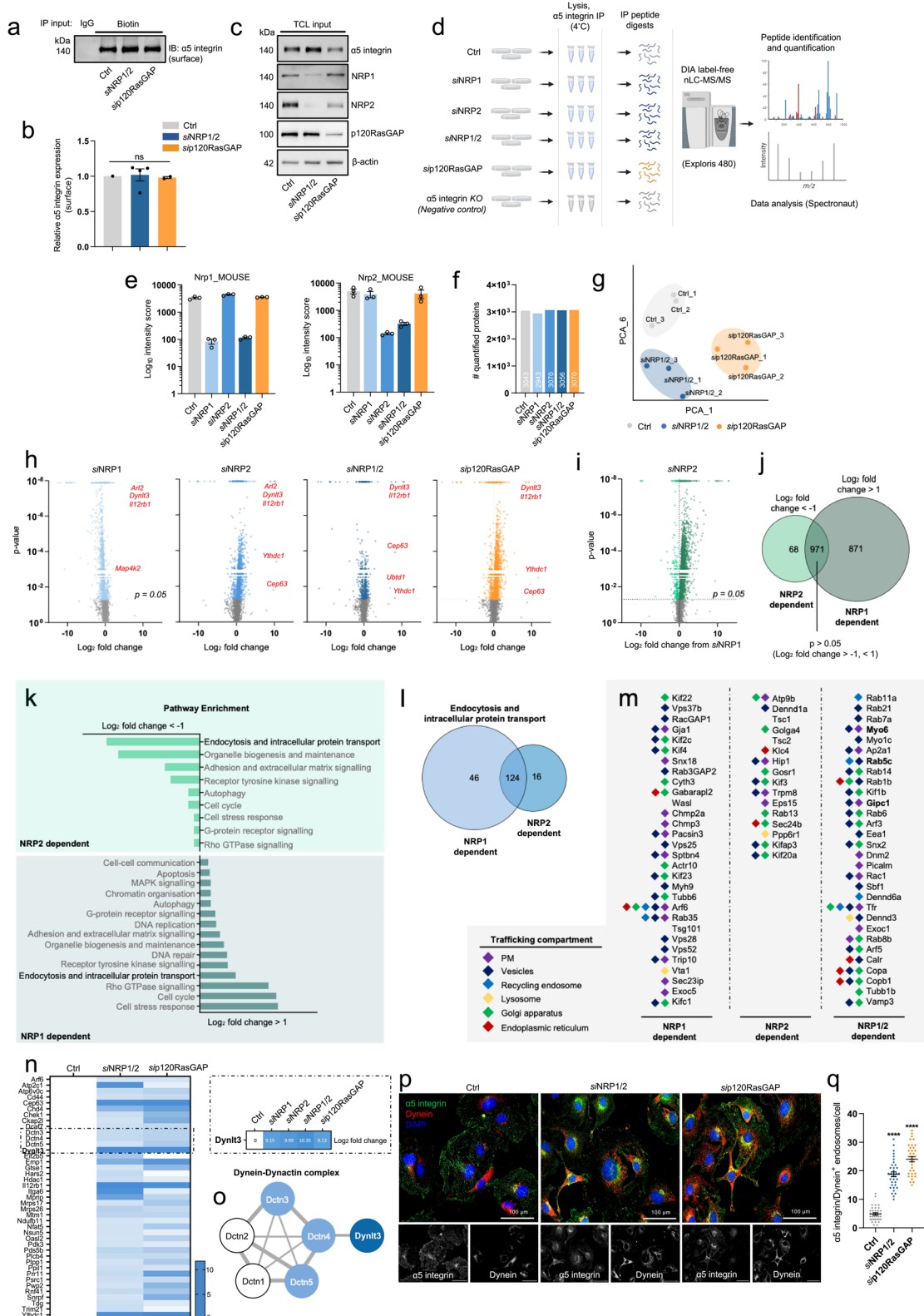

polarisation. Comparative proteomics analysis subsequently identified a candidate escape pathway utilised by *si*NRP1/2 and *si*p120RasGAP ECs to divert α5 integrin away from lysosomal degradation. Importantly, our proteomics workflow enabled us to segregate a comprehensive set of NRP1-dependent and NRP2-dependent α5 integrin interactions, which we invite readers to explore as a resource for further insight.

Previous work has indeed annotated a role for both NRP1 and NRP2 in promoting α5 integrin traffic in ECs[8–10], yet investigations have continued to pursue their roles in isolation rather than to consider their collective function. We observe their robust colocalisation in endocytic punctae at all stages of intracellular traffic, including clathrin-coated assemblies, early endosomes, Rab11 recycling endosomes and Rab7 late endosomes. Indeed,

**Fig. 6 | Retrograde transport via the Dynein-Dynactin complex saves α5 integrin from lysosomal degradation. a** Representative Western blotting showing surface α5 integrin expression from Ctrl, siNRP1/2 and sip120RasGAP lysates. **b** Densitometric quantification of surface α5 integrin expression, ($N \geq 2$ independent experiments), one-way ANOVA + post hoc multiple comparisons tests, ns non-significant. **c** Total cell lysate input showing NRP1, NRP2, α5 integrin and p120RasGAP expression by Western blotting. **d** Schematic representation of DIA nLC-MS/MS workflow from Ctrl and siRNA-depleted lysates. **e** Confirmation of NRP1/NRP2 depletion from MS output. **f** Total numbers of quantified proteins identified in Ctrl, siNRP1, siNRP2, siNRP1/2 and sip120RasGAP samples. **g** PCA plot showing two component separation between Ctrl, siNRP1/2 and sip120RasGAP samples. **h** Volcano plots showing the $\mathrm{Log}_2$ fold-change ratios between siNRP1, siNRP2, siNRP1/2, sip120RasGAP samples and the Ctrl sample. Non-significant ($p < 0.05$) interactions are coloured grey. Protein interactors with the highest positive fold-change are labelled. **i** Volcano plot showing the $\mathrm{Log}_2$ fold-change ratio between siNRP2 and siNRP1 samples. Non-significant ($p < 0.05$) fold-change differences are coloured grey. **j** Venn diagram associated with (**i**) showing the total number of protein

interactors with a $\log_2$ fold-change $<-1$ (NRP$_2$-dependent interactions), the total number of protein interactors with a $\log_2$ fold-change $>1$ (NRP1-dependent interactions), and the total number of protein interactions with a non-significant ($p < 0.05$) fold-change difference. **k** Pathway enrichment analysis showing the top 15 enriched pathways associated with NRP1 and NRP2-specific interactions. **l** Venn diagram associated with (**i**) showing the number of trafficking-associated protein interactors (NRP2 dependent: 16, NRP1 dependent: 46, common: 124. **m** The top 30 protein interactors for each group are shown alongside their location. **n** Heatmap associated with (**h**) showing common interactors between siNRP1/2 and sip120RasGAP samples with a significant, positive $\log_2$ fold-change (>1) away from the Ctrl sample. Highlighted box shows the comparison in $\log_2$ fold-change between all siRNA-depleted groups. **o** STRING interaction network showing associations between Dynlt3 and Dctn1-5. **p** Representative confocal microscopy images showing α5 integrin colocalisation with dynein in Ctrl, siNRP1/2 and sip120RasGAP ECs. **q** Quantification of α5 integrin/dynein$^+$ endosomes/cell, ($n = 30$ cells from three independent biological replicates), Student's $t$-test, ****$p < 0.0001$. **d** Created with BioRender.com.

---

Rab21, EEA1, Rab11 and Rab7 were all identified as common to both NRP1 and NRP2-dependent interactors by our MS analysis. By extension, it is also possible that both NRP receptors are present during the turnover lifecycle of multiple other cell surface complexes, not only α5 integrin. For example, NRP1 has been shown to promote recycling of VEGFR-2 through Rab11$^+$ vesicles, and colocalise with VEGFR-2 in Rab7$^+$ vesicles[53,54]. Given NRP co-silencing impaired α5 integrin distribution to the Rab11 compartment to a greater extent than silencing either NRP receptor individually, we postulate that both share a mutual contribution in the 'long-loop' recycling pathway. For example, only in ECs depleted for both NRP1 and NRP2 did α5 integrin fail to assemble at adhesion sites entirely, and instead accumulate in early and late endosomes.

In addition to reaffirming their shared control of the Rab11 recycling pathway, our MS studies also provided great insight into the NRP's diverging influence on non-canonical α5 integrin traffic in ECs. For example, α5 integrin's interactions with 6 kinesin-family proteins (Kif22, Kif2c, Kif4, Kif23, Kifc1 and Kif2a)[55–59], Sorting nexin 18 (Snx18)[60], and Rab3GAP2, (which acts as a GAP for Rab3[61] and a guanine nucleotide exchange factor (GEF) for Rab18[62]), were all found to be NRP1-specific. Conversely, α5 integrin's interactions with Rab13[63], Golgin A4 (Golga4) and Golgi SNAP receptor complex 1 (Gosr1)[64] were identified as NRP2 dependent. Whilst these NRP-specific interactions should be explored further, we also urge others to consider and investigate NRP-specific modulation of α5 integrin in other cellular pathways, such as the cell cycle and during cell-stress responses, both of which showed an enriched interaction score favouring regulation by NRP1. As such, we propose this α5 integrin interactome dataset as a comprehensive map of candidate associations in ECs.

Despite their reciprocal contribution during Rab11-mediated recycling, we do find it interesting that NRP2 only preferentially associated with p120RasGAP on early endosomes when NRP1 expression is silenced. Furthermore, this interaction only becomes dependent upon FAK phosphorylation in siNRP1 ECs. As p120RasGAP and Rab21 compete for a mutually exclusive interaction with α-integrin tails[17], we hypothesise that NRP1 and NRP2 also competitively associate with p120RasGAP during their transit to Rab11 recycling endosomes. In fibroblasts, fibronectin-integrin-mediated FAK activation was shown to promote SH2-mediated binding of p120RasGAP to p190RhoGAP. In turn, p190RhoGAP phosphorylation stabilised their complex association at leading edge adhesions to regulate cell polarisation[35]. We believe it is likely that internalised integrin-NRP complexes deliver active FAK to p120RasGAP in ECs, thereby promoting polarised integrin recycling and directional migration by facilitating p120-p190 association. For example, sip120RasGAP and siNRP1/2 ECs failed to assemble polarised adhesions and exhibited impaired directional migratory capacity. As p120RasGAP recruitment to membranes has also been connected to a transient attenuation in RhoA signalling and the formation of cell protrusions via upregulated Rac activity in surface adhesions[17,65], it follows that p120RasGAP should be considered as a key regulator of integrin traffic

and signalling to promote cell polarity in ECs. As Rac1 activity was found to be attenuated in ECs depleted for either NRP1 or NRP2[10,66,67], and our MS analyses identified RacGAP1 as an NRP1-specific interactor of α5 integrin, we recommend future studies to pursue this crosstalk further.

Consistent with these conclusions, temporal endothelial-specific deletion of p120RasGAP has been demonstrated to significantly reduce vessel sprouting and filopodia production in the postnatal retina of P6 mice[16], largely phenocopying the defects observed in retinas harvested from NRP1$^{fl/fl}$NRP2$^{fl/fl}$.EC$^{KO}$ animals. In addition to a loss of tip cell polarisation, NRP1$^{fl/fl}$NRP2$^{fl/fl}$.EC$^{KO}$ retinas also exhibited significant reductions in EDA-FN matrix deposition, substantiating our in vitro findings and those reported by Valdembri et al., who described a significant loss of insoluble FN fibres upon NRP1 silencing in human umbilical vein ECs (HUVECs)[8]. Indeed, FN fibrillogenesis relies on the polarised recycling of active α5 integrin[3]. We hypothesise that without stable cell-matrix interactions, NRP1$^{fl/fl}$NRP2$^{fl/fl}$.EC$^{KO}$ vasculature undergoes profound vessel regression at both the vascular front and within the remodelling plexus, resulting in severe vascular sprouting and branching defects.

In summary, we describe a model in which α5 integrin is endocytosed to the early endosome by both NRP1 and NRP2 receptors. The integrin-associated FAK activity mediates the displacement of Rab21 and the shuttling of integrin-NRP complexes to Rab11$^+$ recycling endosomes via a transient but direct interaction with p120RasGAP.

The foremost caveat to any conclusions drawn from these results remains the absence of an in vitro human EC model to consolidate our findings. Whilst beyond the range of this study, it is important that this limitation be considered. For example, whilst the fundamental mechanisms of α5 integrin trafficking are conserved between mouse and human ECs[7–10], expression profiles of NRP1 and NRP2 are likely to differ between (and indeed within) mouse and human endothelial models. For example, even between HUVECs, human microvascular ECs (MECs) and human lymphatic ECs (LECs), the surface density of NRP1 was found to vary markedly and be subject to change dependent on states of VEGF stimulation[68]. Whilst predominantly expressed in the arterial endothelium, NRP2, by contrast, is preferentially expressed on and regulates responses in the lymphatic endothelium[69]. To inform future studies, it would be greatly beneficial to utilise a parameterised computational model similar to that described by Sarabipour et al.[70] (taking into account receptor surface density), to provide mechanistic insight into the differences in trafficking kinetics of such receptors between and within mouse and human endothelial cell lines.

## Methods
### Antibodies and reagents
The following primary antibodies were used for experimental analyses: α5 integrin (CST: 4705S, RRID: AB_2233962, 1:1000; Ab25251, RRID: AB_448738, 1:200), β-actin (CST: 8457S, RRID: AB_10950489, 1:2000), NRP1 (R&D: AF566, RRID: AB_355445, 1:100), NRP2 (SCB: sc-13117,

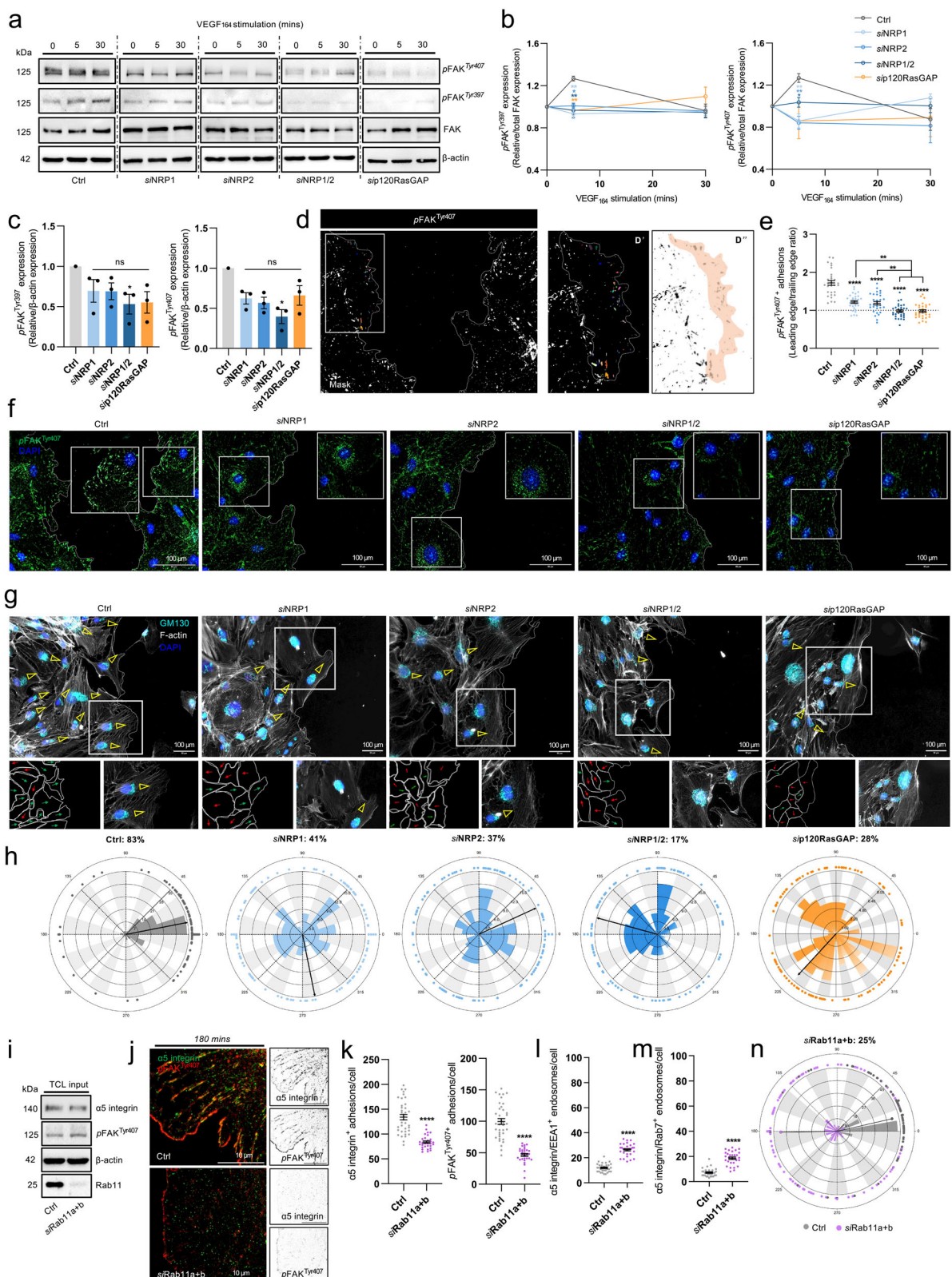

RRID: AB_628044, 1:50; CST: 3366, RRID: AB_2155250, 1:1000), Clathrin heavy chain-1 (Ab21679, RRID: AB_2083165, 1:100), Dynamin-2 (Ab3457, RRID: AB_2093679, 1:100), EEA1 (Ab2900, RRID: AB_2262056, 1:100), Rab11 (Ab3612, RRID: AB_10861613, 1:2000), Rab4 (Ab13252, RRID: AB_2269374, 1:100), Rab7 (CST: 17286, 1:100), αV integrin (Inv: PA5-47096, RRID: AB_2609681, 1:200/1:1000), Paxillin (Ab32084, RRID: AB_779033, 1:100; CST: 2542, RRID: AB_10693603, 1:1000), Tensin-1 (NB: NBP1-84129, RRID: AB_11014791, 1:100), EDA-FN (Sigma: F6140, RRID: AB_476981, 1:400), p120RasGAP (Ab2922, RRID: AB_303418, 1:100/1:1000), Rab21 (SCB: sc-81917, RRID: AB_2253236, 1:100/1:1000), Dynein (Thermo: MA1-070, RRID: AB_2093668, 1:200), FAK (CST: 3285, RRID: AB_2269034, 1:1000), $p$FAK$^{Tyr397}$ (CST: 8556, RRID: AB_10891442,

**Fig. 7 | Endothelial NRPs and p120RasGAP promote cell polarity during directional cell migration. a** Representative Western blotting of Ctrl and siRNA-depleted EC lysates ± 5 or 30 mins VEGF$_{165}$ stimulation, showing expression changes in total FAK, $p$FAK$^{Tyr397}$ and $p$FAK$^{Tyr407}$. **b** Supporting densitometric timecourse analysis of $p$FAK$^{Tyr397}$ and $p$FAK$^{Tyr407}$ expression shown in (**a**), relative to total FAK expression, ($N \geq 3$ independent experiments), one-way ANOVA + post hoc multiple comparisons tests, *$p < 0.05$, **$p < 0.01$. **c** Densitometric analysis of unstimulated $p$FAK$^{Tyr397}$ and $p$FAK$^{Tyr407}$ expression shown in (**a**), relative to β-actin loading control, ($N = 3$ independent experiments), one-sample $t$-test, *$p < 0.05$. **d** Immunolabelling of endogenous $p$FAK$^{Tyr407+}$ leading vs trailing edge focal adhesions the scratch-wound edge. D' and D" panels show the process of skeletonising focal adhesions prior to analysis. **e** Ratio quantification of $p$FAK$^{Tyr407+}$ leading vs trailing edge adhesions in Ctrl and siRNA-depleted ECs at the scratch-wound edge, ($n = 30$). **f** Representative confocal microscopy images quantified in (**e**).

**g** Representative confocal microscopy images showing GM130$^+$ Golgi body positioning and F-actin protrusions in Ctrl and siRNA-depleted ECs at the scratch-wound edge fixed 8-h post scratch initiation. **h** Polar plots showing Golgi body orientation in Ctrl and siRNA-depleted ECs, ($n = 100$). **i** Total cell lysate input confirming siRNA depletion of Rab11 by Western blotting. **j** Representative confocal microscopy images showing α5 integrin colocalisation with $p$FAK$^{Tyr407}$ in Ctrl and *si*Rab11a + b ECs following 180 min adhesion to FN. **k** Quantification of α5 integrin$^+$ and $p$FAK$^{Tyr407+}$ adhesions/cell in Ctrl and *si*Rab11a + b ECs, ($n = 30$ cells from three independent biological replicates), Student's $t$-test, ****$p < 0.0001$. **l** Quantification of α5 integrin$^+$ EEA1 endosomes/cell, ($n = 30$ cells from three independent biological replicates), Student's $t$-test, ****$p < 0.0001$. **m** Quantification of α5 integrin$^+$ Rab7 endosomes/cell, ($n = 30$ cells from three independent biological replicates), Student's $t$-test, ****$p < 0.0001$. **n** Polar plot showing Golgi body orientation in Ctrl and *si*Rab11 ECs, ($n = 100$).

1:1000), $p$FAK$^{Tyr407}$ (Inv: 44-650 G, RRID: AB_2533708, 1:1000/1:200), GM130 (NB: NBP2-53420, RRID: AB_2916095, 1:100), Collagen IV (Ab19808, RRID: AB_445160, 1:500).

CST: Cell Signalling Technologies; R&D: R&D Systems; SCB: Santa Cruz Biotechnology; Ab: Abcam; Inv: Invitrogen; NB: Novus Biologicals

tdTOMATO- expressing α5 integrin constructs were generated by standard PCR protocols using the following primer sequences: 5'-CTATGGGGAGCTGGACGC-3', 5'-GCATCTGAGGTGGCTGGA-3'. The corresponding PCR product was subsequently cloned into linearised tdTOMATO-C1 plasmid (Addgene #54653), before transfected into ECs by electroporation.

### Animal breeding and generation
All experiments were performed in accordance with UK home office regulations and the European Legal Framework for the Protection of Animals used for Scientific Purposes (European Directive 86/609/EEC). All experiments were approved by the Animal Welfare and Ethical Review Board (AWERB) committee at the University of East Anglia, UK. We have complied with all relevant ethical regulations for animal use. NRP1 floxed (NRP1$^{fl/fl}$)[71] and NRP2 floxed (NRP2$^{fl/fl}$)[72] mice generated on a C57/BL6 background were purchased from The Jackson Laboratory (Bar Harbour, Maine, USA). α5 integrin-floxed (α5$^{fl/fl}$) mice (generated on a C57/BL6 background) were kindly provided by Richard Hynes (Massachusetts Institute of Technology, USA). Floxed mice were crossed with tamoxifen-inducible PDGFB.iCreER$^{T2}$ mice, provided by Marcus Fruttiger (UCL, London, UK). Gene deletion was achieved either via tamoxifen administration in vivo or TAT-Cre recombinase nucleofection in vitro.

### Cell isolation, immortalisation and cell culture
Primary mouse lung microvascular endothelial cells (mLMECs) were isolated from age-matched (3–6 weeks) wild-type (WT) or floxed C57/BL6 mice. Cellular digests were expelled through a 19 gauge-needle and filtered through a 70 μm sterile strainer (Fisher Scientific). Cell pellets were seeded onto plasticware coated with a solution of 0.1% gelatin containing 10 μg/ml human plasma fibronectin (FN) (Millipore). and collagen type 1. mLMECs were twice positively selected for using endomucin primary antibody and magnetic activated cell sorting (MACS) as previously described by Reynolds & Hodivala-Dilke[73], prior to immortalisation using polyoma-middle-T-antigen (PyMT) as previously described by Robinson et al.[74].

ECs were cultured in a 1:1 mix of Ham's F-12:Dulbecco's Modified Eagle Medium (DMEM) (low glucose) medium supplemented with 10% foetal bovine serum (FBS), 100 units/mL penicillin/streptomycin (P/S) and 50 μg/mL heparin (Sigma) at 37 °C in a humidified incubator (+5% CO$_2$) unless otherwise stated.

For experimental analyses, plasticware was coated using 10 μg/ml human plasma FN passaging of ECs did not exceed 20. VEGF-stimulation was achieved using 30 ng/ml VEGF-A$_{164}$ (VEGF-A) (murine equivalent to VEGF-A$_{165}$) post 3 h incubation in serum-free medium (OptiMEM®; Invitrogen). VEGF-A was made in-house, as previously described by Krilleke et al.[75].

### TAT-Cre recombinase/siRNA transfection
To excise α5 integrin from ECs isolated from α5 integrin-floxed mice, ECs were twice nucleofected with TAT-Cre recombinase (70 units, Sigma) in the same manner as siRNA oligofection, before being allowed to recover for 24 h.

Trypsinised ECs were subject to oligofection using either non-targeting control siRNA (Ctrl) or murine-specific siRNA duplexes suspended in nucleofection buffer (200 mM Hepes, 137 mM NaCl, 5 mM KCl, 6 mM D-glucose and 7 mM Na$_2$HPO$_4$ in nuclease-free water). Oligofection was performed according to manufacturer's instructions using the Amaxa 4D-nucleofector system (Lonza).

NRP1 siGENOME siRNA duplexes: D-040787-*03*, D-040787-*04*
NRP2 siGENOME siRNA duplexes: D-040423-*03*, D-040423-*04*

### Western blotting
ECs were lysed in lysis buffer (Tris-HCL: 65 mM pH 7.4, sucrose: 60 mM, 3% SDS), homogenised and analysed for protein concentration using the bicinchoninic acid (BCA) assay (Pierce). Equivalent protein concentrations were run on 8% polyacrylamide gels before being subject to SDS-PAGE. Proteins were transferred to a 0.45 μm Amersham Protran® nitrocellulose membrane (GE Healthcare, Amersham) before being incubated in primary antibody resuspended in 5% milk-powder at 4 °C. Membranes were washed with 0.1% Tween-20 in PBS (PBST) and incubated in appropriate horse-radish peroxidase (HRP)-conjugated secondary antibody (Dako) diluted 1:2000 for 2 h at RT. Bands were visualised by incubation with a 1:1 solution of Pierce ECL Western Blotting Substrate (Thermo). Chemiluminescence was detected on a ChemiDoc$^{TM}$ MP Imaging System (BioRad). Densitometric readings of band intensities were obtained using ImageJ$^{TM}$.

### Co-immunoprecipitation assays
ECs were placed on ice before being lysed at 4 °C in lysis buffer (25 mM Tris-HCl, pH 7.4, 100 mM NaCl, 2 mM MgCl$_2$, 1 mM Na$_3$VO$_4$, 0.5 mM EGTA, 1% Triton X-100, 5% glycerol, supplemented with Halt$^{TM}$ protease inhibitors) and cleared by centrifugation at $12,000 \times g$ for 20 min at 4 °C. Supernatant proteins were then quantified using the BCA assay and equivalent protein concentrations (800 μg) were immunoprecipitated with Dynabeads$^{TM}$ Protein G (Invitrogen) coupled to a primary antibody at 4 °C. Immunoprecipitated proteins, alongside a control IgG sample (Fisher Scientific: SA1-200, RRID: AB_325994), were separated by SDS-PAGE and subjected to Western blot analysis.

### Mass spectrometry analysis
Mass spectrometry was carried out by the Fingerprints Proteomics Facility, Dundee University, UK, on EC samples immunoprecipitated against α5 integrin primary antibody as described in 'co-immunoprecipitation assays'. Briefly, IP samples were resolved via 1D SDS-PAGE stained with Quick Coomassie Stain. Gel lanes were then excised and subjected to in-gel processing before overnight trypsin digestion. Digested peptides were run on a Q-ExactivePlus (Thermo) instrument coupled to a Dionex Ultimate 3000 HPLC system. Raw data was acquired in Data Independent Acquisition

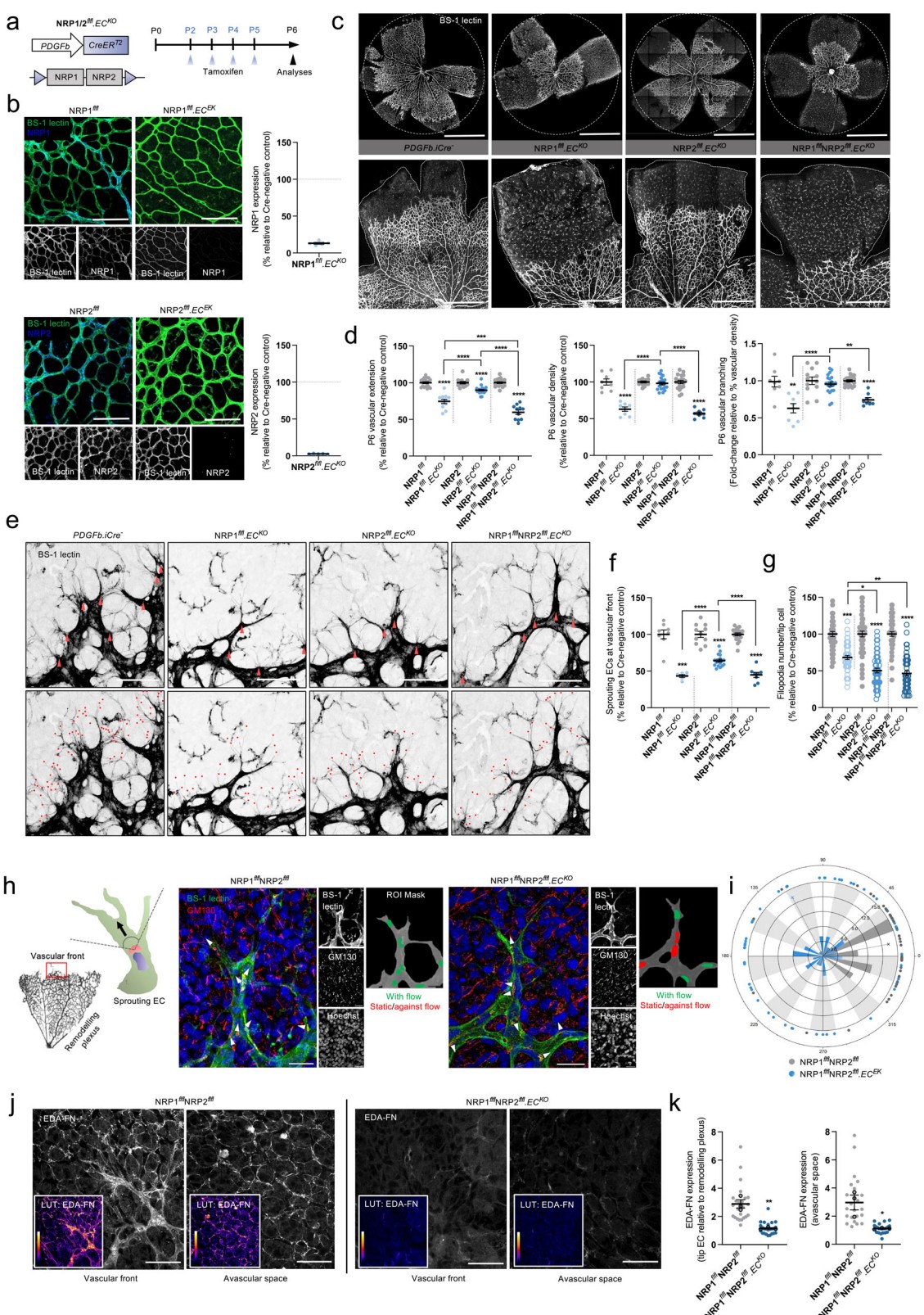

(DIA) mode. A scan cycle comprised a full MS scan with an m/z range of 345–1155, resolution of 70,000, Automatic Gain Control (AGC) target of $3 \times 10^6$ and a maximum injection time of 200 ms. MS scans were followed by DIA scans of dynamic window widths with an overlap of 0.5 Th. DIA spectra were recorded with a first fixed mass of 200 m/z, resolution of 17,500, AGC target $3 \times 10^6$ and a maximum IT of 55 ms. The normalised

collision energy was set to 25% with a default charge state set at 3. Data for both MS scan and MS/MS DIA scan events were acquired in profile mode. Raw data was analysed, and significance was calculated using Spectronaut v17.

STRING interaction networks analysing identified proteins were defined based on default parameters, with the minimum required

**Fig. 8 | Sprouting angiogenesis and FN fibrillogenesis is impaired following NRP depletion in vivo. a** Inducible, endothelial-specific deletion of NRP1, NRP2 or both NRP1 and NRP2 was achieved by crossing NRP$^{fl/fl}$ mice with Pdgfb-iCreER$^{T2}$ mice. Activation of Pdgfb-iCreER$^{T2}$ was achieved by subcutaneous injection of tamoxifen on P2 and P3, followed by intraperitoneal (IP) injections on P4 and P5. Retinas were then harvested at P6. **b** Tamoxifen-induced deletion of NRP1 and NRP2 was confirmed by immuno-colocalisation with BS-1 lectin. **c** Representative confocal microscopy images showing BS-1 lectin-labelled retinal vasculature of Pdgfb-iCreER$^{T2}$-negative, NRP1$^{fl/fl}$.EC$^{KO}$, NRP2$^{fl/fl}$.EC$^{KO}$ and NRP1$^{fl/fl}$NRP2$^{fl/fl}$.EC$^{KO}$ animals. **d** Quantification of retinal vascular extension, vascular density and vascular branching (relative to vascular density), shown as relative percentages of respective littermate control animals, ($N ≥ 3$ independent experiments ($n ≥ 6$ retinas)), Student's $t$-tests/one-way ANOVA + post hoc multiple comparisons tests, **$p < 0.01$, ***$p < 0.001$, ****$p < 0.0001$. **e** Representative confocal microscopy images showing BS-1 lectin-labelled tip cells (red arrows) and filopodia (red dots) in Pdgfb-iCreER$^{T2}$-negative, NRP1$^{fl/fl}$.EC$^{KO}$, NRP2$^{fl/fl}$.EC$^{KO}$ and NRP1$^{fl/fl}$NRP2$^{fl/fl}$.EC$^{KO}$ animals. **f, g** Quantification of the number of sprouting ECs, ($N ≥ 3$ independent experiments ($n ≥ 6$ retinas)), Student's $t$-tests/one-way ANOVA + post hoc multiple

comparisons tests, ***$p < 0.001$, ****$p < 0.0001$ (**f**), and filopodia number/ tip EC (**g**), ($n ≥ 50$ ECs from five retinas), Student's $t$-tests/one-way ANOVA + post hoc multiple comparisons tests, *$p < 0.05$, **$p < 0.01$, ***$p < 0.001$, ****$p < 0.0001$, shown as relative percentages of respective littermate control animals. **h** (Left panel schematic): representation of a sprouting EC showing the correct orientation of the Golgi body (red) in relation to the nucleus (blue) and the direction of migration (arrow). (Right panels): Representative confocal microscopy images of the retinal vascular front immunolabelled with BS-1 lectin, GM130 and Hoechst in Pdgfb-iCreER$^{T2}$ negative and NRP1$^{fl/fl}$NRP2$^{fl/fl}$.EC$^{KO}$ animals. White arrows indicate the direction of Golgi body orientation. **i** Polar plot showing Golgi body orientation in Pdgfb-iCreER$^{T2}$ negative and NRP1$^{fl/fl}$NRP2$^{fl/fl}$.EC$^{KO}$ animals ($n = 50$ ECs). **j** Representative confocal microscopy images showing endogenous EDA-FN at the vascular front and in the avascular space of Pdgfb-iCreER$^{T2}$ negative and NRP1$^{fl/fl}$NRP2$^{fl/fl}$.EC$^{KO}$ animals. **k** (Left): Quantification of EDA-FN expression in sprouting tip cells relative to the remodelling plexus, ($n ≥ 24$ sprouting tip cells from three retinas), (Right): Quantification of avascular EDA-FN expression directly ahead of the leading edge, ($n ≥ 24$ ROIs from three retinas), Student's $t$-tests, *$p < 0.05$, **$p < 0.01$. **h** Created with BioRender.com.

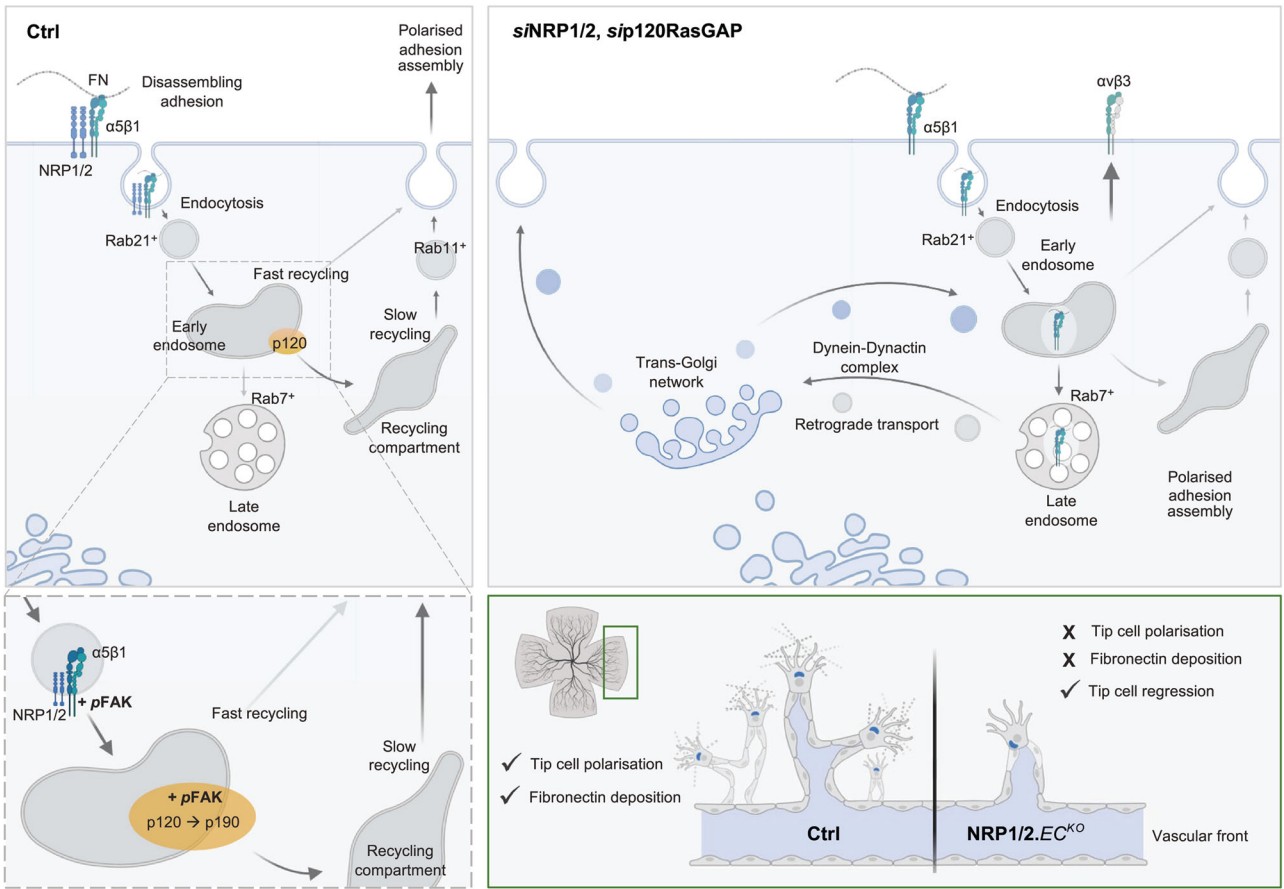

**Fig. 9 | Summary schematic.** Internalised NRP-α5 integrin complexes undergo Rab11$^+$ long-loop recycling via p120RasGAP, promoting polarised nascent adhesion assembly and directional migration in endothelial cells. In the absence of NRP1 and NRP2, α5 integrin arrests in early endosomes and Rab7$^+$ late endosomes, before being recycled via retrograde transport and the dynein–dynactin complex. In vivo, endothelial deletion of NRP1 and NRP2 severely impairs polarised sprouting and fibronectin deposition. Created with BioRender.com.

confidence score of 0.400. Connecting line thickness corresponds to confidence of interaction.

Pathway enrichment analysis was performed using the Reactome database and software[76]. 'Gene expression and maintenance' and Protein and RNA metabolism' pathways are not shown.

**Immunocytochemistry**

ECs were seeded at low density onto pre-coated, acid-washed, sterilised glass coverslips for the indicated timepoints before fixation in 4%

paraformaldehyde (PFA). Fixed ECs were incubated with 10% goat serum in PBS 0.3% triton X-100 prior to primary antibody overnight at 4 °C. Following primary antibody incubation, ECs were incubated with an appropriate Alexa fluor secondary antibody diluted 1:200 before mounting using flouromount-G with DAPI™ (Invitrogen). Images were captured using a Zeiss LSM880 Airyscan Confocal microscope at 63x magnification with an Axiocam 503 mono camera. Adhesion number and size were quantified using ImageJ™ software as previously described by Lambert et al.[77], using an FA size lower detection limit of 0.8 microns. Scratch-wound

immunofluorescence studies were performed by making a scratch through confluent ECs, then allowing ECs to polarise for 8 h prior to PFA fixation.

### RNA extraction and RT-qPCR analysis

Total RNA was extracted using the Promega SV Total RNA Isolation System (#Z3100) from Ctrl and siRNA-depleted ECs. cDNA was prepared using the Promega GoScript™ Reverse Transcriptase Kit (#A5000). RT-qPCR analysis was performed using a Viia7 RT-qPCR System using Promega GoTaq® PCR mastermix (#A6102) following the manufacturer's instructions. The following oligonucleotide primer/probe sets were used: mouse NRP1: Mm00435379_m1; mouse NRP2: Mm00803099_m1; mouse GAPDH: Mm99999915_g1 (Thermo).

### Adhesion assays

ECs were seeded onto 96-well microtiter plates (Costar) pre-coated with FN overnight, then incubated in 5% bovine serum albumin (BSA). ECs were left to adhere for the indicated timepoints before fixation in 4% PFA. Plates were incubated with methylene blue solution, then washed in dH2O prior to de-stain solution (1:1 ethanol:0.1 M HCL). Absorbance was read at 630 nm and normalised to media-only control wells.

### Recycling assays

ECs were placed on ice before being washed in Soerensen buffer (SBS) pH 7.8 (14.7 mM $KH_2PO_4$, 2 mM $Na_2HPO_4$ and 120 mM Sorbitol pH 7.8) as previously described by Remacle et al.[78]. Cell surface proteins were subsequently labelled with 0.3 mg/ml EZ-Link™ Sulfo-NHS-SS-Biotin (Thermo Scientific) at 4 °C. Unreacted biotin was quenched with 100 mM glycine, before biotin-labelled surface proteins were allowed to internalise for 20 min at 37 °C. Following internalisation, ECs were washed with SBS buffer pH 8.2, and incubated with 100 mM sodium 2-mercaptoethane sulfonate (MESNA) at 4 °C. The internalised fraction was then allowed to recycle for the indicated timepoints at 37 °C. Following each timepoint, ECs were again subjected to MESNA incubation at 4 °C, alongside a MESNA-control sample. Excess MESNA was quenched with 100 mM iodoacetamide (Sigma) at 4 °C, prior to a further SBS pH 8.2 wash. ECs were then lysed at 4 °C in lysis buffer (25 mM Tris-HCl, pH 7.4, 100 mM NaCl, 2 mM $MgCl_2$, 1 mM $Na_3VO_4$, 0.5 mM EGTA, 1% Triton X-100, 5% glycerol, supplemented with Halt™ protease inhibitors) and cleared by centrifugation at $12,000 \times g$ for 20 min at 4 °C. Supernatant proteins were then quantified using the BCA assay and equivalent protein concentrations (800 µg) were immunoprecipitated with Dynabeads™ Protein G (Invitrogen) coupled to a mouse anti-biotin primary antibody at 4 °C. Immunoprecipitated biotin-labelled proteins were then separated by SDS-PAGE and subjected to Western blot analysis. The extent of recycled α5 integrin was determined by normalising MESNA-treated samples against their respective internal MESNA⁻ control at each timepoint.

### Retinal angiogenesis assays

Inducible, endothelial-specific deletion of NRP1 and/or NRP2 was achieved by subcutaneous tamoxifen injections (50 µl, 2 mg/ml stock) on postnatal (P) days 2–3, followed by intraperitoneal (IP) injections of the same dose on P4-P5. Mice were sacrificed on P6, and retinas were harvested as previously described. Dissected retinas were fixed in PFA for 30 min before permeabilised in PBS 0.25% triton X-100. Retinas were then incubated in Dako serum-free protein blocking solution for 1 h, before incubated in primary antibody. Following primary antibody incubation, retinas were washed in 0.1% triton X-100 and incubated in the appropriate Alexa fluor secondary antibody before being mounted with Flouromount-G. Images were captured using a Zeiss LSM880 Airyscan Confocal microscope with an Axiocam 503 mono camera. All analyses were performed using ImageJ™ software unless otherwise stated. AngioTool™ software was used to analyse mean vascular density and vessel branching from multiple ROIs per retinal leaflet. The mean number of EC sprouts/retina and filopodia number/sprouting EC were enumerated manually. EDA-FN expression was measured using relative corrected total cell fluorescence (CTCF) values (calculated according to the formula: IntDen - area * background IntDen).

### Statistics and reproducibility

All graphic illustrations and analyses to determine statistical significance were generated using GraphPad Prism 9 software unless otherwise stated. Student's $t$-tests were used to determine statistical significance between Ctrl and an individually shown siRNA group, or between an individual, paired Cre-negative to Cre-positive comparison. Statistical analysis between Ctrl and multiple siRNA groups, or between NRP1$^{fl/fl}$.EC$^{KO}$, NRP2$^{fl/fl}$.EC$^{KO}$, and NRP1$^{fl/fl}$NRP2$^{fl/fl}$.EC$^{KO}$ groups were performed using one-way ANOVA tests with post hoc multiple comparison tests. Bar charts show mean values with standard error of the mean (+SEM). Polarity histogram analyses were performed using NCSS software. Asterisks indicate the statistical significance of $p$ values: $p > 0.05$ ns (not significant), $*p < 0.05$, $**p < 0.01$, $***p < 0.001$ and $****p < 0.0001$.

### Reporting summary

Further information on research design is available in the Nature Portfolio Reporting Summary linked to this article.

## Data availability

The raw data supporting the conclusions of this article will be made available by the authors, without undue reservation, to any qualified researcher. The source data behind the graphs in the paper can be found in Supplementary Data 1. Uncropped and unedited blot/gel images can be found in Supplementary Figs. 7–9. The mass spectrometry proteomics data have been deposited to the ProteomeXchange Consortium via the PRIDE[79] partner repository with the dataset identifier PXD051778.

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

## Acknowledgements

This work was supported by funding from BHF (grant number PG/22/11033); S.D.R. gratefully acknowledges the support of the Biotechnology and Biological Sciences Research Council (BBSRC); this research was partially funded by the BBSRC Institute Strategic Programme Food Microbiome and Health BB/X011054/1 and its constituent project BBS/E/F/000PR13632.

## Author contributions

Conceptualisation: C.J.B. and S.D.R.; Formal analyses and investigation: C.J.B., R.T.J. and J.A.G.E.T.; Resources, supervision and funding acquisition: S.D.R.; Review and editing: C.J.B., R.T.J., J.A.G.E.T., J.R.L. and S.D.R.; Visualisation: C.J.B., R.T.J. and J.A.G.E.T.

## Competing interests

The authors declare no competing interests.
