## [Peer Review File · Communications Biology]

Reviewers' comments:

Reviewer #1 (Remarks to the Author):

Using mouse endothelial cells, Benwell et al investigate the potential redundant roles of NRP1 and NRP2 in the trafficking of $\alpha 5$ integrin; they also explore the endocytic complexes involved. They demonstrate that NRP1 and NRP2 bind to p120RasGAP during the transportation of $\alpha 5$ integrin-FAK complexes, however NRP2 only associates when NRP1 is downregulated. Together, NRP1 and NRP2 coordinate the recycling of $\alpha 5$ integrin through p120RasGAP, facilitating polarized endothelial sprouting angiogenesis over fibronectin matrices.

The manuscript provides interesting new insights into the neuropilin-dependent $\alpha 5$ integrin trafficking, based on proteomic analysis and extensive imagery. Thus, the manuscript is of general interest, but a few questions need to be addressed.

Main points:

- In this study, the in vitro model is based on mouse endothelial cells. The level of expression of NRP1 and NRP2 might differ between endothelial models. We encourage the authors to consolidate their main data with at least one human EC model. In this line, NRP1 level seems downregulated upon NRP2 siRNA (Fig 1), the authors should address by qPCR whether it is at RNA level or due to cross-reactivity of the antibodies. Also, fibronectin coating was used throughout the manuscript. Experiments with more complex matrices/surfaces should be devised. These are important points.
- Some conclusions are not supported by the results: the stated high colocalization between NRP1 and NRP2 (Fig 1K-M). Moreover, the lysosomal degradation was not experimentally challenged with inhibitors (Fig 7). Same hold true for the transport via the Dynein-Dynactin complex of $\alpha 5$ integrin, which needs to be further investigated beyond mass-spectrometry analysis.
- Proximity-ligation assays should be performed to reinforce the conclusions on NRPs-p120RasGAP interactions.
- The total versus plasma membrane localization is barely supported by experimental data (biotin assay). The authors should explore with flow cytometry and/or staining typically used to discriminate between internalized and membrane staining (37°C incubation, acid wash ...). Active and/or blocking antibodies can also be useful.
- Data quantifications would strengthen the conclusions based on image analysis: Figures 1-K, 1-L, 4-A.
- Ig control antibodies for ImmunoPrecipitation experiments should be added.
- The authors should consider to improve the clarity of their manuscript both with text editing (long sentences for multiple results/approaches and panels) and figure organization (for instance: Figures 2E right panel and 2F confirms the result from 2E left panel; Figure 5K Rab4 staining; Figures 6B-C and Suppl. Figure 5A are repetitive analysis; Figure 6D; Figure 8I is the overlap of Suppl. Figure 6F, they show

the same result). This would ease the reading. Moreover, the authors should clarify better what is behind the differences between the adhesion phenotype in light of their results (NRP1 alone, NRP2 alone, and NRP1 and NRP2 siRNA).

Minor points:

- Some Figures are not cited or discussed in the results (Suppl. Figure 1C-D; Figure 5D)
- Some panels are fuzzy or too small: Figure 1E, 2C-D, 6-H, 6-N, 7-H, 8-I
- Rab7 staining is missing (used for the quantification in Figure 3H).
- Figure 4I-K: the cells from sip120RasGAP condition seem smaller compared to others. Please check the scales. Along with this line, immunofluorescence scale values are too small.
- Please improve the quality of image panels. too dark (ex: Figure 1H; Figure 3F) and not clear (EEA1 red staining in Figure 4K).
- Reference needed for 'Both NRP1 and NRP2 have been independently identified as key regulators of $\alpha 5$ integrin trafficking and fibronectin fibrillogenesis in EC'.
- Please edit the text in few places:
 - In the abstract: 'comprative'.
 - In the legend of Figure 2C-D: 'adhesion size (μM)' should be (μm^2)
 - Figure 2I: indication in the microscopy images 'siNRP1/2' instead of 'siNRP2'.
 - Figure 3F: there is a letter 'E' in the middle of the Ctrl merge image.
 - Figure 6C: 'pFAKTyr397 and Tyr407' in the main text.
 - In the text, 'defective translocation of $\alpha 5$ integrin from fibrillar adhesions to EEA1+ early endosomes (Figure F-G)' is missing the indication of the number 4.
 - In the text, 'Next, by means of biochemical immunoprecipitation assays (Suppl. Figure 3B), ... (Suppl. Figure 3C)' and 'First, we silenced Rab21 expression (to inhibit internalisation) (Suppl. Figure 3D)' should be number Suppl. Figure 4.

Reviewer #2 (Remarks to the Author):

The paper by Benwell et al. presents a thorough analysis of alpha 5 integrin trafficking, defining the importance of NRP-1/2 and the associated redundancy. The authors use confocal microscopy and proteomic approaches alongside trafficking assays to define the role of key proteins including p120RasGAP. A loss of integrin alpha 5 from surface adhesions inhibited endothelial cell polarisation and impairs fibronectin deposition and polarised sprouting.

The findings are novel and provide an important refinement of established pathways. Some of the data presentation is over complicated and confusing and makes it more difficult to interpret.

Issues across multiple figures

- The size of scale bars used in confocal images was absent (they may be on the actual bar but they are too small to read).
- Where number of cells are reported (e.g. in colocalization studies) this is reported in legends as (in the

case of Fig 1Q) “n=30”. This should be changed to “n=30, from x biological replicates/animals”. Statistics should be performed across experimental replicates and consider using an alternative statistical test, t-tests become less reliable with n>30

- Some of the confocal imagery in the paper requires additional post-processing – some panels are too dim, even in grayscale, particularly at their current size. On a related note, the use of bright yellow in graphs in e.g. Fig 1J/I should be changed as it is nearly impossible to identify against a white background (presumably why the authors used a black background in the image annotations).
- Log10 is often used incorrectly– the authors appear to be using non log values on a log10 axis rather than transforming the values and plotting on a linear axis. For example, in Fig1C the authors state that the cutoff p value is 0.01. Accordingly, the values begin at 10⁻². However, log₁₀(0.01) is -2, thus the graph, and the more routinely used -log₁₀(0.01) is 2. Oddly Log₂ is used correctly in later figures.
- The authors report mass spec data with an adj.p-value. It is not stated how these values are FDR-adjusted (e.g Benjamini-Hochberg).
- Statistical tests should be reported in situ, ideally in figure legends. The authors must also report the post hoc tests accompanying their use of ANOVAs.
- The use of different shades, rather than different colours to represent e.g. siNRP1 vs siNRP2 vs siNRP1/2 makes some graphs very hard to interpret. I appreciate it allows the authors to maintain a single colour for a particular group across multiple figures, but it’s very damaging to some individual figures.
- The reporting of statistical comparisons is very difficult to interpret at times, often using asterixis above the data point for treatment vs control and bracket comparisons between treatments (see Fig 1Q). This could perhaps be harmonised to solely brackets, and the authors must take care to include all comparisons including those which are non-significant – e.g. in Fig 1Q I am assuming they performed the siNRP1 vs siNRP1/2 comparison and it was non-significant.

In detail

Fig.1 and associated text

- In Fig 1.C the proteomics as displayed are unclear. Is there a rationale for picking out NRP1/2 etc. it appears they are cherry picked. Do the other hits make sense and do they support the thesis of the paper or agree with other published literature.
- “Next, we visualised the spatial segregation between endogenous NRP1 and NRP2 in endocytic compartments where $\alpha 5$ integrin has been reported to reside in” – authors should provide citations and/or images to support this (in supp. Figs would be sufficient).
- The small images below Fig.1H are not helpful consider removing them or at least increase the contrast.
- In Fig. I the clathrin labelling appears adjacent to the TGN- presumably this is not endocytic clathrin recruited by AP-2?
- Fig1M includes the line upon which the intensity was analysed where other panels instances do not. This line should be included on all panels which use this method of assessing colocalization.
- Are the images to assess colocalization max intensity projections or single planes- for co-localisation; the latter is preferable.

Fig 2 and associated text

- An explanation of EDA FN would be useful for the wider audience.

- C/D (and M) possibly due to a combination of the points on graph, colours and stats this is hard to understand. Whilst it's generally good to include all data points, here it might be worth using only the averages or perhaps a violin plot, or including expanded versions in the supplementals? Or at least re-sizing, re-colouring and re-working the statistical comparisons.
- E and L– whilst I appreciate the need for the gray value x distance graphs for four colour images; when there are only two channels I think something like Mander's co-localisation coefficient should be used to give a quantitative values.
- I images are captioned siNRP2, when this should be siNRP1/2?

Fig.3

This is generally convincing in terms of phenotype.

Fig.4 and associated text

- The same image is used in 4D as in 1P and this is the same quantification. It should be clearly stated that this is a repeat of the previous data and I'm not sure it's necessary to repeat the graph of the quantification. Presumably the experiment has been repeated a number of times could another representative image be used? This happens in Fig 5.J, Fig 6E.
- “confirmed the defective translocation of $\alpha 5$ integrin from fibrillar adhesions to EEA1+ early endosomes (Figure F-G)” Figure number missing
- A – the colocalization should be quantified as previously done (gray value x distance)
- B – it's not made clear enough this is a surface biotinylation assay based on the text. Representative blots should also be included in the supplementary data, including biotin blots that confirm successful cleavage at each step and the lack of penetrance of the biotin (via a lack of biotinylated actin in pulldown)

Fig 5 and associated text

- The way the pull downs and the inputs are presented is confusing. Are they from the same experiment, they need to be to verify the levels of receptor in the input? This needs to be clarified.
- It is also appears from the legend that only the siNRP1 + NRP2 IP pulldown was repeated (accordingly it is the only one with a satisfactory representative image).
- To make the associated conclusions all the blots should be powered and quantified
- The discussion of NRP1 and NRP2 competing for $\alpha 5$ integrin binding is tricky. The authors are technically correct in describing it as such but it is hard to differentiate between competition and compensation here – particularly as the authors identify different binding proteins for the two but cannot identify different outcomes for the binding (or different physiological circumstances where it might be relevant).
- The Rab7+ co-staining indicating a shunt to lysosomal traffic doesn't tally well with previous assertions that total protein levels are unchanged when this recycling pathway is disrupted. I believe the authors recognise this, given they later investigate the possibility of the dynein-dynactin pathway, but their logic is not readily apparent to the reader initially. It might be better to recognise this and pose it (and the NRP1/2 competition point) as unanswered questions.
- Directly following from these points, I suggest that Figures 6 and 7 could be swapped, so that the discussion of different binding partners and the possibly dynein-dynactin contribution follows directly

from Figure 5. The article would then examine functional contributions of this pathway in vitro (current Figure 6) and finally in vivo (Figure 8).

Fig 6 and associated text

- B – axes should be the same on both graphs. I also believe all the samples should be normalised to basal Ctrl samples (or at least both sets of normalisation should be presented (either in supps or as a replacement for C). I understand the viewpoint that the idea is to show the change in pFAK (which is clearer if each group is normalised to its own basal level) but would argue that the pFAK levels and change relative to a control cell(s) are also physiologically relevant information.
- C – as stated in general points, adding ns labels would be beneficial here
- G – I would recommend arrows showing the direction of mis-aligned Golgi in a different colour (similar to what was done in the in vivo data) as this would make these images more impactful.

Fig 7 and associated text

- “Cep63, Dynlt3 and Il12rβ1 were all found to interact with α5 integrin at a similarly high magnitude, indicating their reciprocal contribution in the absence of the NRPs or p120RasGAP” Given the lack of a secondary validation of this (e.g. via IF) and that it’s also arguably the case for Ythdc1 (which is not involved in this putative pathway); I find this and other presentations of the putative dynein dynactin link too strongly worded. Either the validation is needed or the language needs toning down slightly.
- E- Rather than showing individual bar charts these should perhaps be combined with individual points and error.
- “Log2 fold-change < -1 (which we defined as NRP2-dependent interactions), 871 associated at a Log2 fold-change > 1” whilst log2 fold-change is used accurately, I feel like using the raw fold change values here (and throughout this section) is far more beneficial for the reader’s understanding - in this specific case a <0.5 fold decrease and >2 fold increase is more digestible.
- O I feel like this proposed pathway model should be expanded (e.g. to include pFAK and the functional outcomes) and included at the very end of the article to summarise it, rather than here. Or have a functional summary figure in Fig 8 and a molecular one here. The integrins and NRPs should be visible throughout the pathways in both panels. Finally, the image needs to be of higher resolution.

Fig. 8

- This figure is convincing and nicely pulls together the data.

Discussion

- As mentioned earlier, some discussion of different physiological circumstances where the different NRP1/NRP2 trafficking partners might be relevant would be appreciated. Potentially the differing roles of FN/integrins in the lymphatic vs vascular systems, where different NRPs dominate?
- “We believe it is likely that internalised integrin-NRP complexes deliver active FAK to p120RasGAP in ECs, thereby promoting polarised integrin recycling and directional migration by facilitating p120-p190 association” a summary model showing this would improve the paper.

Generally this paper represents a significant amount of work and overall the conclusions are convincing. Some effort to make the figures consistent and easier to digest would greatly improve the article.

Typos

In the abstract it says comprative label- rather than comparative..

Reviewers' comments:

Reviewer #1 (Remarks to the Author):

In this study, the in vitro model is based on mouse endothelial cells. The level of expression of NRP1 and NRP2 might differ between endothelial models. We encourage the authors to consolidate their main data with at least one human EC model.

- *We appreciate the reviewers comment here and agree that validating the findings presented in this manuscript would be beneficial, however would argue that it falls beyond the scope of this study within the timeframe required. We have published multiple times using these mouse endothelial cells and have validated their behaviour against primary counterparts.*
- *We have now included a limitations section at the end of the discussion (page 9, lines 37-49) commenting on the benefits of validating our results in human cells and discussing the need to take into account varying expression profiles of surface receptors in different endothelial models.*

In this line, NRP1 level seems downregulated upon NRP2 siRNA (Fig 1), the authors should address by qPCR whether it is at RNA level or due to cross-reactivity of the antibodies.

- *We now present R-qPCR (Taqman) analysis (Suppl. Figure 1F) (page 3 lines 4-5; page 19 lines 29-32) showing there are no gene changes following siRNA silencing. As the reviewer suggests, the small protein change we observe by western blotting may be due to antibody cross reactivity, but we feel that this does not affect the outcome of any results.*

Also, fibronectin coating was used throughout the manuscript. Experiments with more complex matrices/surfaces should be devised. These are important points.

- *We believe that the reviewer does not fully understand our utility of fibronectin matrices for these studies. Fibronectin is used in isolation to specifically isolate $\alpha 5$ integrin adhesion dynamics. We would therefore argue that a more complex matrix coating would hinder our ability to dissect the roles of NRP1 and NRP2 in regulating $\alpha 5$ integrin adhesion and traffic. Furthermore, fibronectin is by far the most dominant matrix component in the mouse retina, in which we validate our findings in Figure 8.*

Some conclusions are not supported by the results: the stated high colocalization between NRP1 and NRP2 (Fig 1K-M).

- *We have now uploaded a high-resolution jpeg of this figure, which hopefully demonstrates a strong colocalisation between NRP1 and NRP2, particularly in trafficking vesicles.*

Moreover, the lysosomal degradation was not experimentally challenged with inhibitors (Fig 7).

- *We are slightly confused by this comment and the need to experimentally challenge with an inhibitor. We show that $\alpha 5$ integrin **does not** undergo lysosomal degradation, but instead accumulates in Rab7 endosomes, prior to being re-directed by retrograde transport. As such, challenging with a lysosomal inhibitor in this case would be unlikely to show any change as total expression levels do not change.*

Same hold true for the transport via the Dynein-Dynactin complex of $\alpha 5$ integrin, which needs to be further investigated beyond mass-spectrometry analysis.

- *We have now validated our mass spectrometry analysis with immunofluorescence imaging and analysis of dynein colocalising with $\alpha 5$ integrin in Ctrl, siNRP1/2 and sip120RasGAP ECs, now shown in Figure 6P-Q/ page 6 (lines 35-37).*

Proximity-ligation assays should be performed to reinforce the conclusions on NRPs-p120RasGAP interactions.

- *We agree with the reviewer here and believe that proximity-ligation assays would greatly reinforce the conclusions drawn regarding the interactions between the NRPs and p120RasGAP. Unfortunately, we have been unable to find any p120RasGAP antibody adequate enough to use for such PLA studies, and as such have been limited to showing their interactions by Co-IP and IF studies.*

The total versus plasma membrane localization is barely supported by experimental data (biotin assay). The authors should explore with flow cytometry and/or staining typically used to discriminate between internalized and membrane staining (37°C incubation, acid wash ...). Active and/or blocking antibodies can also be useful.

- *We would argue here that our biotin studies clearly show that $\alpha 5$ integrin surface expression is unchanged following co-depletion of NRP1 and NRP2 or depletion of p120RasGAP. Unfortunately, $\alpha 5$ integrin is trypsin sensitive, preventing us from reliably measuring its localisation by flow cytometry. Equally, we would argue that measuring total and surface levels biochemically by biotin labelling and western blotting is more quantitative and reliable than by membrane staining and IF imaging. We believe that the blots shown in Figure 2G, Figure 6A and Suppl. Figure 4D quantitatively show membrane vs total expression of $\alpha 5$ integrin*

Data quantifications would strengthen the conclusions based on image analysis: Figures 1K, 1L, 4A.

- *Colocalisation gray value x distance maps have now been added and are now presented in Figure 1N and Figure 4B respectively.*

Ig control antibodies for ImmunoPrecipitation experiments should be added.

- *IgG lanes for all immunoprecipitation experiments are now shown as requested.*

The authors should consider to improve the clarity of their manuscript both with text editing (long sentences for multiple results/approaches and panels) and figure organization (for instance: Figures 2E right panel and 2F confirms the result from 2E left panel; Figure 5K Rab4 staining; Figures 6B-C and Suppl. Figure 5A are repetitive analysis; Figure 6D; Figure 8I is the overlap of Suppl. Figure 6F, they show the same result). This would ease the reading.

- *We would argue here that all figures presented are necessary to make our conclusions, and all are associated with relevant analysis:*
 - *Whilst Figures 2F confirms the results shown in 2E, we do so with an $\alpha 5$ integrin-Tdtomato expression construct rather than a primary antibody. Expression constructs offer greater specificity than certain polyclonal antibodies, and therefore we felt this was necessary to validate our findings in this example.*
 - *Figure 5F Rab4 staining (now presented in Figure 5I-J): we believe confirming no changes to $\alpha 5$ integrin's localisation in the Rab4 compartment after PF562271 treatment was necessary to complement our findings presented in Figure 1N,*

showing that the NRPs do not colocalise with this trafficking endosome and therefore likely do not regulate its transport of internalised cargoes.

- Figures 6B-C, Suppl. Figure 5A (now presented in Figure 7B-C, Suppl. Figures 5A-B): whilst we agree in premise that these analyses are repeated, each graph has been shown to better understand the complexity of the data presented. Figure 7B is used to show the mean fold-change in pFAK expression from each group's basal 0-minute condition, normalised to total FAK expression. Figure 7C is used to demonstrate that under basal 0-minute conditions, pFAK expression differs between Ctrl and siRNA groups. Suppl. Figures 5A-B show all data points presented in Figure 7B across all timepoints, normalised to total FAK expression or β -actin loading control. Whilst Suppl. Figure 5A shows all points normalised to each group's respective basal 0-minute condition, Suppl. Figure 5B shows all points normalised to the Ctrl group's basal 0-minute condition. These supplemental figures should be used only as a reference point to understand the variation shown in Figure 7B-C.
- Data shown in Suppl. Figure 6F is indeed an overlap of Figure 8I and was included only to ease inspection of the results.

Minor points:

Some Figures are not cited or discussed in the results (Suppl. Figure 1C-D; Figure 5D)

- Figures are now cited on page 2 (line 35) and page 5 (line 10) respectively.

Some panels are fuzzy or too small: Figure 1E, 2C-D, 6-H, 6-N, 7-H, 8-I

- We have now uploaded high-resolution jpegs which hopefully resolves this issue across figures.

Rab7 staining is missing (used for the quantification in Figure 3H).

- Rab7 staining images have now been added and are shown in Figure 3H-I.

Figure 4I-K: the cells from sip120RasGAP condition seem smaller compared to others. Please check the scales. Along with this line, immunofluorescence scale values are too small.

- Larger scale values have been added to resolve this issue on all IF images.

Please improve the quality of image panels. too dark (ex: Figure 1H; Figure 3F) and not clear (EEA1 red staining in Figure 4K).

- We have now uploaded high-resolution jpegs which hopefully resolves this issue across figures.

Reference needed for 'Both NRP1 and NRP2 have been independently identified as key regulators of α 5 integrin trafficking and fibronectin fibrillogenesis in EC'.

- References added on page 2 (lines 18-19).

Please edit the text in few places:

In the abstract: 'comprative'.

- Text updated on page 1 (line 18).

In the legend of Figure 2C-D: 'adhesion size (μ M)' should be (μ m²)

- Text updated on page 19 (line 38).

Figure 2I: indication in the microscopy images 'siNRP1/2' instead of 'siNRP2'.

- *Text updated in Figure 2I.*

Figure 3F: there is a letter 'E' in the middle of the Ctrl merge image.

- *Letter 'E' removed in Figure 3F.*

Figure 6C: 'pFAKTyr397 and Tyr407' in the main text.

- *Updated in 6C (now Figure 7C) to correct misused FAK residue.*

In the text, 'defective translocation of $\alpha 5$ integrin from fibrillar adhesions to EEA1+ early endosomes (Figure F-G)' is missing the indication of the number 4.

- *Updated to Figure 4G-H on page 4 (line 42).*

In the text, 'Next, by means of biochemical immunoprecipitation assays (Suppl. Figure 3B), ... (Suppl. Figure 3C)' and 'First, we silenced Rab21 expression (to inhibit internalisation) (Suppl. Figure 3D)' should be number Suppl. Figure 4.

- *Text updated on page 4 (lines 31, 32, 33, 35, 41) respectively.*

Reviewer #2 (Remarks to the Author):

Issues across multiple figures

The size of scale bars used in confocal images was absent (they may be on the actual bar but they are too small to read).

- *Larger scale values have been added to resolve this issue on all IF images.*

Where number of cells are reported (e.g. in colocalization studies) this is reported in legends as (in the case of Fig 1Q) "n=30". This should be changed to "n=30, from x biological replicates/animals". Statistics should be performed across experimental replicates and consider using an alternative statistical test, t-tests become less reliable with $n > 30$

- *This notation has now been updated on all figure legends to include biological replicates. All statistics comparing multiple groups/conditions have now been updated to one-way ANOVA tests with Post Hoc multiple comparison tests.*

Some of the confocal imagery in the paper requires additional post-processing – some panels are too dim, even in grayscale, particularly at their current size. On a related note, the use of bright yellow in graphs in e.g. Fig 1J/I should be changed as it is nearly impossible to identify against a white background (presumably why the authors used a black background in the image annotations).

- *We have now uploaded high-resolution jpegs which hopefully resolves this issue across figures. We have also placed all gray value x distance graphs over a darker background to improve clarity.*

Log10 is often used incorrectly– the authors appear to be using non log values on a log10 axis rather than transforming the values and plotting on a linear axis. For example, in Fig1C the authors state that the cutoff p value is 0.01. Accordingly, the values begin at 10^{-2} . However, $\log_{10}(0.01)$ is -2, thus the graph, and the more routinely used $-\log_{10}(0.01)$ is 2. Oddly Log2 is used correctly in later figures.

- *The values presented in Figure 1C have now been Log10 transformed as suggested, and the axes updated.*

The authors report mass spec data with an adj.p-value. It is not stated how these values are FDR-adjusted (e.g Benjamini-Hochberg).

- *Apologies, this is a mistake on our part and to clarify, no adjustment (other than being log10 transformed) was performed on any p values. Figures have now been updated to reflect this and figure axes now read 'log10(p value)' only.*

Statistical tests should be reported in situ, ideally in figure legends. The authors must also report the post hoc tests accompanying their use of ANOVAs.

- *All statistical tests are now reported in situ in the respective figure legends as requested.*

The use of different shades, rather than different colours to represent e.g. siNRP1 vs siNRP2 vs siNRP1/2 makes some graphs very hard to interpret. I appreciate it allows the authors to maintain a single colour for a particular group across multiple figures, but it's very damaging to some individual figures.

- *We have now uploaded high-resolution jpegs which hopefully resolves this issue across figures.*

The reporting of statistical comparisons is very difficult to interpret at times, often using asterixis above the data point for treatment vs control and bracket comparisons between treatments (see Fig 1Q). This could perhaps be harmonised to solely brackets, and the authors must take care to include all comparisons including those which are non-significant – e.g. in Fig 1Q I am assuming they performed the siNRP1 vs siNRP1/2 comparison and it was non-significant.

- *We have now included and present all relevant non -significant comparisons, for example on Figure 1R, Figure 5J, Suppl. Figure 2B, Suppl. Figure 3B and Suppl. Figure 6E. Not all non-significant comparisons are presented as we believe it would negatively affect the figures and add unnecessary complexity.*

In detail:

Fig.1 and associated text

In Fig 1.C the proteomics as displayed are unclear. Is there a rationale for picking out NRP1/2 etc. it appears they are cherry picked. Do the other hits make sense and do they support the thesis of the paper or agree with other published literature.

- *We believe the reviewer has misunderstood the premise of performing proteomics here- we and others have independently published a role for NRP1 and NRP2 individually regulating integrin traffic. This manuscript discusses their cooperative function. This proteomics analysis was initially performed to isolate $\alpha 5$ integrin-specific binding partners, of which we highlight both NRP1 and NRP2 in Figure 1C, alongside $\alpha 5$ integrin, $\beta 1$ integrin and fibronectin ($\alpha 5 \beta 1$ integrin-specific matrix). The other relevant hits are discussed and validated in the text on page 2 (lines 23-35), and in Suppl. Figure 1.*

“Next, we visualised the spatial segregation between endogenous NRP1 and NRP2 in endocytic compartments where $\alpha 5$ integrin has been reported to reside in” – authors should provide citations and/or images to support this (in supp. Figs would be sufficient).

- *Citations have now been included on page 2 (line 48).*

The small images below Fig.1H are not helpful consider removing them or at least increase the contrast.

- *We have now uploaded high-resolution jpegs which hopefully resolves this issue across figures.*

In Fig. I the clathrin labelling appears adjacent to the TGN- presumably this is not endocytic clathrin recruited by AP-2?

- *Clathrin is known to localise at the plasma membrane and around the TGN, mediating transport from the cell surface to sorting endosomes and the Golgi via its AP2 adaptor. Equally, the clathrin shown in Figure 1I may also be linked to its AP1 adaptor, mediating transport from the TGN to endosomes or promoting intra-Golgi recycling of secretory cargoes.*

Fig1M includes the line upon which the intensity was analysed where other panels instances do not. This line should be included on all panels which use this method of assessing colocalization.

- *Lines on colocalisation images have now been added as requested.*

Are the images to assess colocalization max intensity projections or single planes- for co-localisation; the latter is preferable.

- *To clarify, all images to assess colocalisation are single plane images.*

Fig 2 and associated text

An explanation of EDA FN would be useful for the wider audience.

- *An explanation of EDA-FN has now been provided on page 4 (line 3).*

C/D (and M) possibly due to a combination of the points on graph, colours and stats this is hard to understand. Whilst it's generally good to include all data points, here it might be worth using only the averages or perhaps a violin plot, or including expanded versions in the supplementals? Or at least re-sizing, re-colouring and re-working the statistical comparisons.

- *Figures 2C and 2D have now been changed to only show data from unstimulated cells. As such the graphs have been resized and include a violin plot overlay. The full datasets for these figures have now been moved to Suppl. Figure 2A.*

E and L- whilst I appreciate the need for the gray value x distance graphs for four colour images; when there are only two channels I think something like Mander's co-localisation coefficient should be used to give a quantitative values.

- *Mander's colocalisation coefficients have now been included for Figures 2E and L as requested.*

I images are captioned siNRP2, when this should be siNRP1/2?

- *Apologies, this has now been updated accordingly.*

Fig.3

This is generally convincing in terms of phenotype.

Fig.4 and associated text

The same image is used in 4D as in 1P and this is the same quantification. It should be clearly stated that this is a repeat of the previous data and I'm not sure it's necessary to repeat the graph of the quantification. Presumably the experiment has been repeated a number of times could another representative image be used? This happens in Fig 5.J, Fig 6E.

- *This was an oversight on our part. Alternative representative images have now been included for Figure 4D (now presented in Figure 4E). The associated analyses for Figure 4E and Figure 5I now only includes the relevant data groups.*
- *We are unsure why the reviewer mentions Figure 6E here (now Figure 7E). Figure 7E shows new data regarding leading vs trailing edge pFAK⁺ adhesions and has not been presented previously in the manuscript figures.*

“confirmed the defective translocation of $\alpha 5$ integrin from fibrillar adhesions to EEA1+ early endosomes (Figure F-G)” Figure number missing

- *Text updated on page 4 (line 42), now referring to Figure 4G-H).*

A – the colocalization should be quantified as previously done (gray value x distance)

- *Colocalisation gray value x distance maps have now been included alongside Figure 4A (Maps shown in 4B).*

B – it's not made clear enough this is a surface biotinylation assay based on the text.

- *The text has been updated to clearly state this is a surface biotinylation study (page 4, line 31-32).*

Representative blots should also be included in the supplementary data, including biotin blots that confirm successful cleavage at each step and the lack of penetrance of the biotin (via a lack of biotinylated actin in pulldown).

- *We have now included a representative biotin recycling blot in Suppl. Figure 4D as requested.*

Fig 5 and associated text

The way the pull downs and the inputs are presented is confusing. Are they from the same experiment, they need to be to verify the levels of receptor in the input? This needs to be clarified. It also appears from the legend that only the siNRP1 + NRP2 IP pulldown was repeated (accordingly it is the only one with a satisfactory representative image). To make the associated conclusions all the blots should be powered and quantified.

- *Apologies for any confusion caused. Yes, the inputs shown in Figure 5D are from the same experiment as Figure 5A. The figures have now been reorganised so that the TCL inputs are directly following Figure 5A and are now shown in Figure 5B. Figure 5C shows a repeated image of siNRP1 + NRP2 IP pulldown followed by Western blotting for p120RasGAP, and Figure 5D shows the associated quantification. No differences were observed for the other pulldown groups, and therefore quantifications were not presented.*

The discussion of NRP1 and NRP2 competing for $\alpha 5$ integrin binding is tricky. The authors are technically correct in describing it as such but it is hard to differentiate between competition and compensation here – particularly as the authors identify different binding proteins for the two but cannot identify different outcomes for the binding (or different physiological circumstances where it might be relevant).

The Rab7+ co-staining indicating a shunt to lysosomal traffic doesn't tally well with previous assertions that total protein levels are unchanged when this recycling pathway is disrupted. I believe the authors recognise this, given they later investigate the possibility of the dynein-dynactin pathway, but their logic is not readily apparent to the reader initially. It might be better to recognise this and pose it (and the NRP1/2 competition point) as unanswered questions.

Directly following from these points, I suggest that Figures 6 and 7 could be swapped, so that the discussion of different binding partners and the possibly dynein-dynactin contribution follows directly from Figure 5. The article would then examine functional contributions of this pathway in vitro (current Figure 6) and finally in vivo (Figure 8).

- *We appreciate the reviewer's comment here and have switched Figures 6 and 7 accordingly with the associated text changes to ensure continuity.*

Fig 6 and associated text (now updated as Figure 7)

B – axes should be the same on both graphs. I also believe all the samples should be normalised to basal Ctrl samples (or at least both sets of normalisation should be presented (either in supps or as a replacement for C). I understand the viewpoint that the idea is to show the change in pFAK (which is clearer if each group is normalised to its own basal level) but would argue that the pFAK levels and change relative to a control cell(s) are also physiologically relevant information.

- *The axes for these graphs have now been changed accordingly, and an additional set of normalised data is now presented in Suppl. Figure 5B as requested.*

C – as stated in general points, adding ns labels would be beneficial here

- *NS labels have now been added as requested.*

G – I would recommend arrows showing the direction of mis-aligned Golgi in a different colour (similar to what was done in the in vivo data) as this would make these images more impactful.

- *We have already included red vs green arrows to assign mis-aligned vs aligned Golgi in this Figure (bottom panels).*

Fig 7 and associated text (now updated as Figure 6)

“Cep63, Dynlt3 and Il12rβ1 were all found to interact with α5 integrin at a similarly high magnitude, indicating their reciprocal contribution in the absence of the NRPs or p120RasGAP” Given the lack of a secondary validation of this (e.g. via IF) and that it's also arguably the case for Ythdc1 (which is not involved in this putative pathway); I find this and other presentations of the putative dynein dynactin link too strongly worded. Either the validation is needed or the language needs toning down slightly.

- *We have now validated our mass spectrometry analysis with immunofluorescence imaging and analysis of dynein colocalising with α5 integrin in Ctrl, siNRP1/2 and sip120RasGAP ECs, now shown in Figure 6P-Q/ page 6 (lines 35-37).*

E- Rather than showing individual bar charts these should perhaps be combined with individual points and error.

- *Figure E has now been updated as requested to show combined points with error bars.*

“Log2 fold-change < -1 (which we defined as NRP2-dependent interactions), 871 associated at a Log2 fold-change > 1” whilst log2 fold-change is used accurately, I feel like using the raw fold change values here (and throughout this section) is far more beneficial for the reader's understanding - in this specific case a <0.5 fold decrease and >2 fold increase is more digestible.

- *We recognize the reviewers point here, and have now included a text change on page 6 (line 18) to clarify the difference between Log2 fold-change > 8 and actual fold-change to ease understanding. The majority of published mass spec datasets are presented as Log2 values. Furthermore, we felt that presenting Log2 transformed data in the figures but linear values in the text would be confusing.*

○ I feel like this proposed pathway model should be expanded (e.g. to include pFAK and the functional outcomes) and included at the very end of the article to summarise it, rather than here. Or have a functional summary figure in Fig 8 and a molecular one here. The integrins and NRPs should be visible throughout the pathways in both panels. Finally, the image needs to be of higher resolution.

- *We have now uploaded high-resolution jpegs which hopefully resolves this issue across figures. An expanded pathway schematic has now also been included as requested and will now be included as Figure 9. Text updated accordingly on page 8 (line 20), and on page 23 (lines 27-32).*

Fig. 8

This figure is convincing and nicely pulls together the data.

Discussion

As mentioned earlier, some discussion of different physiological circumstances where the different NRP1/NRP2 trafficking partners might be relevant would be appreciated. Potentially the differing roles of FN/integrins in the lymphatic vs vascular systems, where different NRPs dominate?

- *An additional paragraph has now been added to the discussion on page 9 (lines 37-49) commenting on these points.*

“We believe it is likely that internalised integrin-NRP complexes deliver active FAK to p120RasGAP in ECs, thereby promoting polarised integrin recycling and directional migration by facilitating p120-p190 association” a summary model showing this would improve the paper.

- *An expanded pathway schematic has now also been included as requested and will now be included as Figure 9. Text updated accordingly on page 8 (line 20), and on page 23 (lines 27-32).*

Generally, this paper represents a significant amount of work and overall, the conclusions are convincing. Some effort to make the figures consistent and easier to digest would greatly improve the article.

Typos

In the abstract it says comparative label- rather than comparative..

- *Text updated on page 1 (line 18).*

REVIEWERS' COMMENTS:

Reviewer #1 (Remarks to the Author):

The authors had revised the manuscript and addressed the concerns mainly in their rebuttal.

Please note that there is a typo in Fig 3A label (tenisin instead of tensin)

Reviewer #2 (Remarks to the Author):

Almost all my reservations have been addressed and this is a very interesting manuscript worthy of publication.

I have 2 minor comments:

If the clathrin being referred to in Fig. 1I is at the Golgi as is suggested by its localisation it shouldn't be referred to as a clathrin pit, its more likely a budding coat. The text should be amended-line 101 to remove mention of pit.

In Fig. 7B there are some additional lines between the 2 graphs, this is either an error or an issue with the merged file.

REVIEWERS' COMMENTS:

Reviewer #1 (Remarks to the Author):

The authors had revised the manuscript and addressed the concerns mainly in their rebuttal.

Please note that there is a typo in Fig 3A label (tenisin instead of tensin)

This has been corrected.

Reviewer #2 (Remarks to the Author):

Almost all my reservations have been addressed and this is a very interesting manuscript worthy of publication.

I have 2 minor comments:

If the clathrin being referred to in Fig. 11 is at the Golgi as is suggested by its localisation it shouldn't be referred to as a clathrin pit, its more likely a budding coat. The text should be amended-line 101 to remove mention of pit.

Amended as suggested.

In Fig. 7B there are some additional lines between the 2 graphs, this is either an error or an issue with the merged file.

This has been corrected.